# Cholesteryl ester transfer protein (CETP) as a drug target for cardiovascular disease

Amand F. Schmidt [1,2,3✉], Nicholas B. Hunt [4], Maria Gordillo-Marañón [1,2], Pimphen Charoen [1,5,6], Fotios Drenos[1,7], Mika Kivimaki [8], Deborah A. Lawlor [9,10,11], Claudia Giambartolomei [12], Olia Papacosta[13], Nishi Chaturvedi [1,14], Joshua C. Bis [15], Christopher J. O'Donnell [16,17], Goya Wannamethee[13], Andrew Wong [14], Jackie F. Price[18], Alun D. Hughes [1,2,14], Tom R. Gaunt [9,10,11], Nora Franceschini[19], Dennis O. Mook-Kanamori[20], Magdalena Zwierzyna [1,2], Reecha Sofat[21], Aroon D. Hingorani [1,2,22,23] & Chris Finan [1,2,3,22,23]

Development of cholesteryl ester transfer protein (CETP) inhibitors for coronary heart disease (CHD) has yet to deliver licensed medicines. To distinguish compound from drug target failure, we compared evidence from clinical trials and drug target Mendelian randomization of CETP protein concentration, comparing this to Mendelian randomization of proprotein convertase subtilisin/kexin type 9 (PCSK9). We show that previous failures of CETP inhibitors are likely compound related, as illustrated by significant degrees of between-compound heterogeneity in effects on lipids, blood pressure, and clinical outcomes observed in trials. On-target CETP inhibition, assessed through Mendelian randomization, is expected to reduce the risk of CHD, heart failure, diabetes, and chronic kidney disease, while increasing the risk of age-related macular degeneration. In contrast, lower PCSK9 concentration is anticipated to decrease the risk of CHD, heart failure, atrial fibrillation, chronic kidney disease, multiple sclerosis, and stroke, while potentially increasing the risk of Alzheimer's disease and asthma. Due to distinct effects on lipoprotein metabolite profiles, joint inhibition of CETP and PCSK9 may provide added benefit. In conclusion, we provide genetic evidence that CETP is an effective target for CHD prevention but with a potential on-target adverse effect on age-related macular degeneration.

[1] Institute of Cardiovascular Science, Faculty of Population Health, University College London, London, UK. [2] UCL British Heart Foundation Research Accelerator, London, UK. [3] Department of Cardiology, Division Heart and Lungs, University Medical Center Utrecht, Utrecht, The Netherlands. [4] Division of Pharmacoepidemiology and Clinical Pharmacology, Utrecht Institute for Pharmaceutical Sciences (UIPS), Utrecht University, Utrecht, The Netherlands. [5] Department of Tropical Hygiene, Faculty of Tropical Medicine, Mahidol University, Bangkok, Thailand. [6] Integrative Computational BioScience (ICBS) Center, Mahidol University, Bangkok, Thailand. [7] Department of Life Sciences, College of Health, Medicine, and Life Sciences, Brunel University London, Uxbridge, UK. [8] Department of Epidemiology and Public Health, University College London, London, UK. [9] MRC Integrative Epidemiology Unit at the University of Bristol, Bristol, UK. [10] Population Health, Bristol Medical School, University of Bristol, Bristol, UK. [11] Bristol NIHR Bristol Biomedical Research Centre, University Hospitals Bristol National Health Service Foundation Trust and University of Bristol, Bristol, UK. [12] Istituto Italiano di Tecnologia, Central RNA Lab, Genova, Italy. [13] Primary Care and Population Health, University College London, London, UK. [14] MRC Unit for Lifelong Health and Ageing at UCL, London, UK. [15] Cardiovascular Health Research Unit, Department of Medicine, University of Washington, Seattle, WA, USA. [16] Department of Medicine, Brigham and Women's Hospital, Harvard Medical School, Boston, MA, USA. [17] Department of Medicine, VA Boston Healthcare System, Boston, MA, USA. [18] Usher Institute, University of Edinburgh, Edinburgh, UK. [19] Department of Epidemiology, Gillings School of Global Public Health, University of North Carolina, Chapel Hill, NC, USA. [20] Department of Clinical Epidemiology, Leiden University Medical Center, Leiden, The Netherlands. [21] Institute of Health Informatics, University College London, London, UK. [22] Health Data Research UK, London, UK. [23] These authors contributed equally: Aroon D. Hingorani, Chris Finan. ✉email: amand.schmidt@ucl.ac.uk

The causal role of low-density lipoprotein cholesterol (LDL-C) in coronary heart disease (CHD) has been established through randomized controlled trials (RCTs) of different LDL-C lowering drug classes[1–4] and by Mendelian randomization (MR) studies[5].

Circulating high-density lipoprotein cholesterol (HDL-C) shows an inverse association with CHD in non-randomized studies[6]. MR studies utilizing genetic variants associated with HDL-C selected from throughout the genome have provided inconclusive evidence on the causal role of HDL-C as a biomarker[5,7]. Findings from RCTs of niacin[8] and cholesteryl ester transfer protein (CETP) inhibitors[9], developed to prevent CHD by raising HDL-C have also been disappointing. For example, of the four CETP inhibitors that have progressed to phase 3 clinical trials, none have received market authorization. Two CETP inhibitors (Supplementary Table 1) are still in active development, raising important questions about the validity of CETP as a therapeutic target[10]. One interpretation is that HDL-C is not causally related to CHD and that therefore raising HDL-C as a therapeutic strategy will be an ineffective approach for CHD prevention. As a result, the reduction in CHD events observed in a large RCT of anacetrapib (odds ratio [OR] 0.91, 95% CI: 0.85–0.97)[11], was attributed to its effect on LDL-C rather than to its HDL-C raising action[10].

However, analysis of lipoprotein sub-classes measured using nuclear magnetic resonance (NMR) spectroscopy suggests that, unlike LDL-C, HDL-C encompasses the cholesterol content of several lipoprotein sub-fractions that have differential associations with CHD: some fractions being associated with higher and others with lower CHD risk[12,13]. Failures of CETP inhibitors might be in fact related to the developed compounds rather than the drug target itself either because of inadequate target engagement or competing off-target action. Compound-related failures can be addressed by developing improved CETP inhibitors, whereas target failure would affect *all* CETP inhibitors.

To address these uncertainties, we performed a drug target MR study of CETP, focusing on variants within and around the encoding gene (acting in *cis*) that are associated with circulating CETP concentration, to directly model the effects of pharmacological action on this target by a clean drug with no off-target actions. To evaluate potentially diverse effects of drug target perturbation, we combined drug target MR with a phenome-wide scan of over 190 disease biomarkers or clinical end-points relevant to cardiovascular as well as non-cardiovascular outcomes. Where possible, we compared drug target MR effect estimates to compound-specific effect estimates derived from a systematic review and meta-analysis of CETP inhibitor RCTs. On-target failures would result in consistent treatment effects across all compounds, which should be directionally concordant to the on-target effect modeled through MR, whereas inconsistencies in effects would reflect compound failures due to either inadequate target engagement or off-target effects. Finally, drug target MR analyses of CETP and PCSK9 (an archetypal LDL-C lowering drug target) were compared for their effects on the same outcomes to differentiate anticipated effects of CETP versus PCSK9 inhibition.

Here, we show that previous failures of CETP-inhibitor drugs are likely compound rather than target related. Through drug target MR we show that on-target CETP inhibition is predicted to reduce the risk of several cardiovascular endpoints and diabetes but potentially increase the risk of age-related macular disease (AMD). Comparison with anticipated effects of PCSK9 inhibition suggests that inhibiting both targets might provide added benefit.

## Results

**Effects of different CETP inhibitors in trials**. A systematic review of available literature identified 15 CETP-inhibitor RCTs with at least 24 weeks of follow-up, including four different compounds (six anacetrapib, four dalcetrapib, four torcetrapib, and one evacetrapib study), all evaluated against placebo (Supplementary Table 2) and involving 79,961 participants. Participants received either torcetrapib 60–120 mg, evacetrapib 130 mg, anacetrapib 100 mg, or dalcetrapib 600–900 mg per day, reflecting differences in compound potency (Supplementary Table 2, and Supplementary Note 2). The longest median follow-up times were: 49 months for anacetrapib in the REVEAL trial, 31 months for dalcetrapib in the DAL-OUTCOMES trial, 24 months for torcetrapib in the RADIANCE 1 and ILLUSTRATE trials, and 26 months for evacetrapib ACCELERATE trial.

All four compounds increased HDL-C and reduced LDL-C, but the magnitude of effect differed between compounds (Fig. 1, Supplemental Table 3). Anacetrapib and evacetrapib had the largest HDL-C increasing effect, 130% (95% CI: 127–133) and 132% (95% CI: 130–133) respectively, followed by torcetrapib 52% (95% CI: 49–55) and dalcetrapib 29% (95% CI: 23–43); heterogeneity $p$-value < 0.001. The reduction in LDL-C was −38% (95% CI: −40 to −36) for anacetrapib, −37% (95% CI: −38 to −36) for evacetrapib, −20% (95% CI: −24 to −17) for torcetrapib, and −1% (95% CI: −1.1 to −0.9) for dalcetrapib. The CETP inhibitor effects were similarly heterogenous (interaction $p$-values < 0.001) for triglycerides (TG), apolipoprotein A1, B, lp(a), and systolic/diastolic blood pressure (SBP/DBP); Fig. 1 and Supplemental Fig. 1 and Table 3.

CETP inhibitors also differed in their effects on clinical outcomes (Fig. 1). Torcetrapib increased risk of all-cause mortality (ACM) (OR 1.56, 95% CI: 1.14–2.12), while evacetrapib decreased ACM (OR 0.84, 95% CI: 0.71–1.00); heterogeneity $p$-value = 0.009. Similarly, torcetrapib increased any cardiovascular disease (CVD, OR 1.22, 95% CI: 1.08–1.38), while anacetrapib decreased CVD (OR 0.93, 95% CI: 0.87–1.00); heterogeneity $p$-value 0.002. Anacetrapib reduced any myocardial infarction (MI) risk (OR 0.89, 95% CI: 0.80–0.99), with the remaining compounds showing a neutral MI effect; heterogeneity $p$- value 0.046.

Small study heterogeneity was explored using funnel plots (Supplemental Fig. 2), which did not provide convincing evidence of differential CVD treatment effects by study size; although the number of available studies was limited. Given that the REVEAL trial of anacetrapib only showed treatment benefit after the first 2 years of follow-up[11], we analyzed short follow-up studies separately: OR 0.80 (95% CI: 0.43–1.48) for CVD, and compared this to REVEAL study CVD estimate: OR 0.93 (95% CI: 0.87–1.00), finding no significant difference. Due to the limited number of studies available for each compound, meta-regression analyses failed to provide insights into whether the compound-specific CVD effects depended on baseline characteristics or lipid effects (Supplemental Table 4).

**On-target effects of CETP inhibition using drug target MR.** Genetic instruments for CETP concentration were sourced from a *cis* window (Chr 16: bp: 56,961,923–56,985,845; GRCh38) using aggregated data from Blauw et al.[14]. Lower instrumented CETP concentration was associated with lower LDL-C −0.08 (mmol/L, 95% CI: −0.08 to −0.07), TG −0.09 (mmol/L, 95% CI: −0.10 to −0.08), Lp[a] −2.20 (nmol/L, 95% CI: −2.70 to −1.71), apolipoprotein B −0.03 (g/L, 95% CI: −0.03 to −0.03), and higher HDL-C 0.23 (mmol/L, 95% CI: 0.23–0.24), and apolipoprotein A1 0.13 (g/L, 95% CI: 0.13–0.13); Fig. 2 (full details in Supplementary Table 5).

Lower instrumented CETP concentration was significantly associated with lower blood pressure (−0.21 mmHg for SBP and −0.12 for DBP), lower concentration of blood glucose

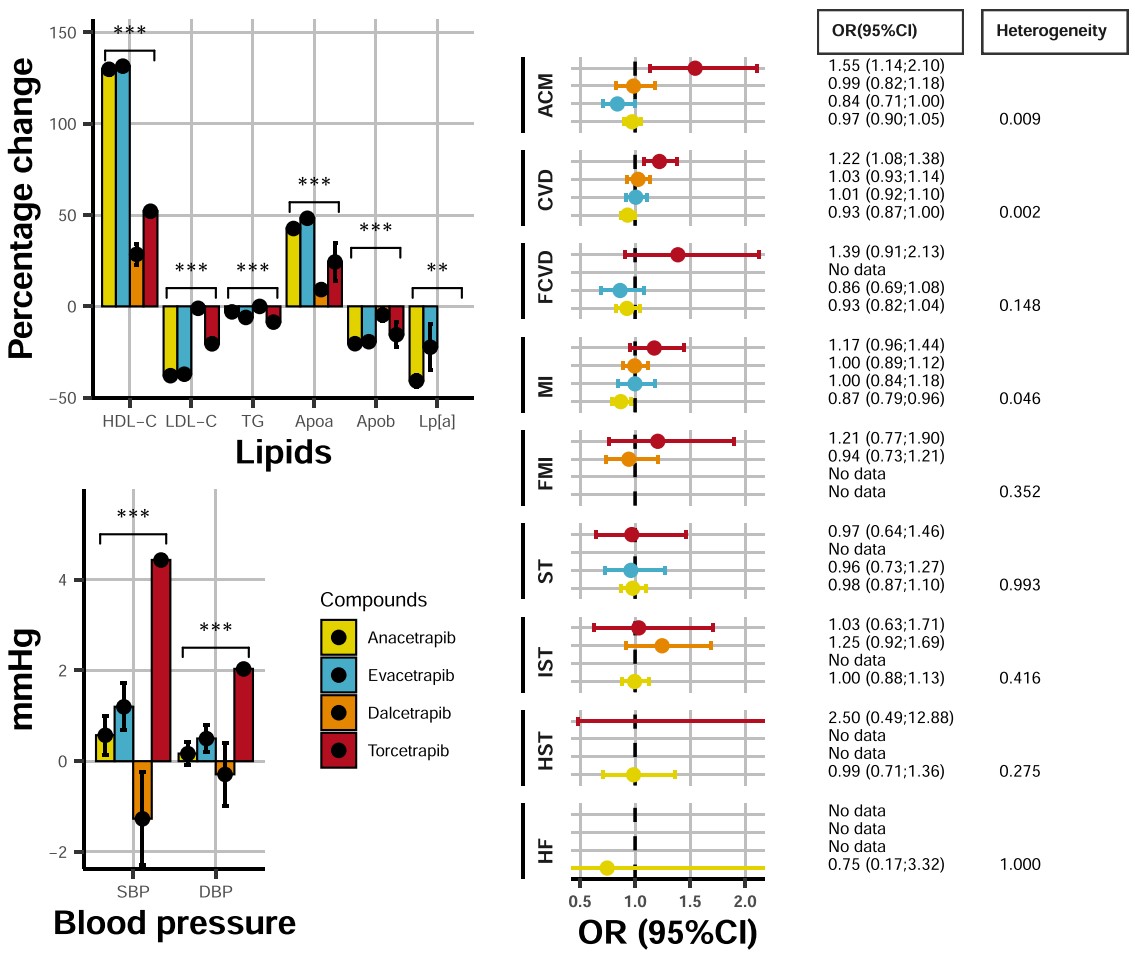

**Fig. 1 Differences in CETP-inhibitor effects on lipids, blood pressure, and clinical endpoints.** N.B Results are based on a fixed effect compound-specific meta-analyses with differences between compounds tested using a Q-test (Heterogeneity). *** indicates a p-value < 0.001 for a two-sided Cochrane's Q-test, without adjustment for multiplicity. The p-values are provided, to 2 dp, in Supplemental Fig. 3 as "Heterogeneity p-value". LDL: LDL-C, HDL: HDL-C, TG: triglycerides, ApoA1: apolipoprotein A1, ApoB: apolipoprotein B, S/DBP: systolic/diastolic blood pressure, ACM: all-cause mortality, CVD: cardiovascular disease, FCVD: fatal-CVD, MI myocardial infarction, FMI: fatal-MI, ST: any stroke, IST: Ischemic stroke, HST: hemorrhagic stroke, HF: heart failure. Error bars reflect 95% confidence intervals, and the central dot represents the odds ratio (RHS) or mean difference (LHS). The total number of subjects and events used in each analysis are provided in Supplemental Fig. 3.

(−0.02 mmol/L), HbA1c (−0.09 mmol/mol), lower cell counts for leukocytes (−0.03 × 10⁹ cells/L), lymphocytes (−0.02 × 10⁹ cells/L), and monocytes (−0.01 × 10⁹ cells/L). These findings were directionally consistent in *cis*-MR analysis weighted by the genetic associations with either LDL-C, HDL-C, or TG acting as proxies for protein concentration and activity (Supplemental Fig. 3).

Lower instrumented CETP concentration was associated with lower risk of CHD (OR 0.95, 95% CI: 0.91–0.99), heart failure (HF) (OR 0.96, 95% CI: 0.93–0.98), CKD (OR 0.94, 95% CI: 0.91–0.97), and higher risk of AMD (OR 1.31, 95% CI: 1.22–1.39); Fig. 3 and Supplemental Table 6. The magnitude and effect direction of the protein concentration (pQTL) weighted analysis were consistent with the LDL-C, HDL-C, and TG weighted analyses (Supplemental Fig. 3, Table 6), which additionally suggested lower CETP protected against type 2 diabetes (T2DM) incidence.

**On-target effects of PCSK9 inhibition using drug target MR.** We compared the drug target MR results of CETP lowering to those for PCSK9, using genetic instruments on PCSK9 concentration (Methods). Lower PCSK9 concentration (Fig. 2, Supplemental Table 5) was associated with lower LDL-C (−0.57 mmol/L),

apolipoprotein B (−0.15 mmol/L), Lp[a] (−3.54 nmol/L), and HDL-C (−0.03 mmol/L). We additionally observed an association with carotid intima-media thickness (−0.02 mm), SBP (−1.20 mmHg), blood urea nitrogen (BUN: −0.04 mg/dl), HbA1c (−0.25 mmol/mol), and higher estimated-GFR (eGFR: 0.01 per SD), C-reactive protein (CRP: 0.41 mg/L), pulse rate (1.22 bpm), and blood cell counts (Fig. 2, and Supplemental Table 5).

Lower PCSK9 concentration was significantly associated with the following clinical endpoints (Fig. 3, Supplementary Table 6): CHD (OR 0.69, 95% CI: 0.59–0.81), any stroke (OR 0.79, 95% CI: 0.69–0.91), any ischemic stroke (OR 0.86, 95% CI: 0.76–0.97), large artery stroke (OR 0.64, 95% CI: 0.47–0.87), HF (OR 0.79, 95% CI: 0.71–0.87), AF (OR 0.90, 95% CI: 0.83–0.97), CKD (OR 0.83, 95% CI: 0.72–0.94), MS (OR 0.69 95% CI: 0.50–0.96), and increased risk of asthma (OR 1.97, 95% CI: 1.56–2.48) and AD (OR 2.43, 95% CI: 1.93–3.06). An LDL-C weighted analysis was consistent with these findings (Fig. 2, Supplemental Fig. 2 and Table 6).

**Lipoprotein subfraction profiles based on NMR spectroscopy.** The drug target MR analysis of NMR assayed metabolites revealed that lower CETP concentration was associated with a wide-ranging number of lipoprotein sub-fraction size and content measures,

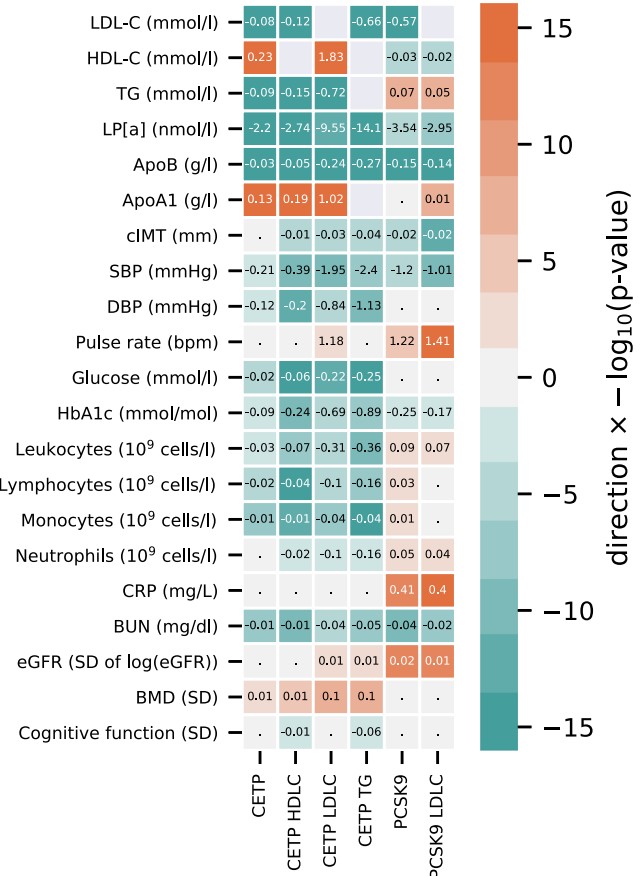

**Fig. 2 Drug target Mendelian randomization estimates of lower CETP and PCSK9 weighted by genetic associations with protein concentration or downstream lipid.** N.B The rows represent the quantitative outcomes and the columns represent the intermediate variable (approximating) drug target concentration. Cells are colored by effect direction times $-\log_{10}(p\text{-value})$, with the mean difference (the slope coefficient), provide for MR results with a $p$-value smaller than 0.05. The $p$-values were truncated at $10^{-16}$ ensuring sufficient variation in the color code. $p$-values were calculated from two-sided $Z$-test statistics, without multiplicity correction. Effects were oriented toward the canonical drug target effect direction: decreasing for CETP concentration, PCSK9 concentration, LDL-C, and for TG, and increasing for HDL-C.

including medium, large, and extra-large HDL-C subfractions, and lower extra-small, small, and medium VLDL sub-fractions (Fig. 4). Lower CETP concentration had a minimal effect on total LDL-C assayed using NMR spectroscopy: 0.00 SD (95% CI: −0.03 to 0.03) for LDL-C, compared to an HDL-C effect of 0.51 SD (95% CI: 0.47–0.54). Lower CETP was however strongly associated with decreased mean LDL-C diameter −0.20 SD (95% CI: −0.17 to −0.23), and to IDL subfractions (Fig. 4). The PCSK9 NMR profile was narrower compared to CETP, with lower PCSK9 associated predominantly with lower medium and large LDL-C subfractions, IDL, and extra small VLDL.

**Comparing effects of CETP inhibitors to drug target MR effects of CETP modulation.** Anacetrapib and evacetrapib displayed a similar risk factor profile that most closely reflected the on-target association of lower CETP concentration modeled genetically and hence clustered most closely to on-target CETP modulation (Fig. 5). However, both torcetrapib and dalcetrapib showed biomarker profiles distinct from that of genetically instrumented lower CETP concentrations. For torcetrapib this

difference was driven by an increasing effect on SBP and DBP. For dalcetrapib this difference was due to attenuated lipid associations.

**MVMR to evaluate lipid mediating pathways.** The drug target MR analyses described above used genetic associations with protein concentration, as well as genetic associations with downstream proxies of protein concentration and activity such as LDL-C or HDL-C. Such analyses provide insight on the effects of protein inhibition, but not necessarily on the mediating pathway. To assess mediation, we employed MVMR to jointly model the effects of these lipid pathways, leveraging genetic associations with NMR assayed measurements which also included Apo-B. The MVMR model for LDL-C and HDL-C (Fig. 6 and Supplemental Table 7) indicated that the CHD decreasing effects of PCSK9 were convincingly mediated by lower LDL-C (OR per SD decrease in LDL-C: 0.66, 95% CI: 0.58–0.75), for CETP we found evidence for HDL-C mediation instead (OR per SD increase in HDL-C: 0.85, 95% CI: 0.82–0.88). These analyses suggest the risk-increasing effect of CETP on AMD was likely due to its HDL-C increasing effect, while the Alzheimer effect of PCSK9 was likely mediated by LDL-C (Fig. 6). Supplanting LDL-C by genetic associations with Apo-B, we observed suggestive but insufficiently precise, evidence of Apo-B mediating the CETP effect on CHD OR 0.60 per SD decrease in Apo-B (95% CI: 0.34–1.03), independent of HDL-C; Supplemental Fig. 4 and Table 8. Finally, both MVMR models for CETP indicate its T2DM protective effect acts likely through HDL-C, independent of either LDL-C or Apo-B.

## Discussion
We found substantial heterogeneity in the effects of four CETP-inhibitors (anacetrapib, evacetrapib, dalcetrapib, and torcetrapib) on major lipid fractions, blood pressure, all-cause mortality, and cardiovascular outcomes, suggesting between-compound differences in the efficacy of CETP inhibition, off-target actions, or both. The profile of anacetrapib and evacetrapib on blood lipids and cardiovascular endpoints most closely matched the effects of genetically instrumented reductions in CETP concentration suggesting that anacetrapib and evacetrapib are effective CETP inhibitors. We note that torcetrapib, and to a maller degree anacetrapib and evacetrapib increased SBP and DBP. This was directionally discordant to the drug target MR effect, where higher concentrations of CETP decreased blood pressure, which was similar to the observed dalcetrapib decreasing effect on SBP.

The reduction in cardiovascular events seen in the REVEAL trial of anacetrapib (median follow-up 1497 days; Supplementary Table 2) is consistent with the drug target MR results presented here. The manufacturer, Merck, did not seek marketing authorization for this drug, citing an anticipated lack of regulatory support[15]. The evacetrapib ACCELERATE trial was terminated for futility after a median follow-up of 791 days, a time point before the benefits of anacetrapib emerged in the REVEAL trial (see Fig. 1 of ref. [11]); the anacetrapib effect on major coronary events increased to a rate ratio of 0.80 (95% CI: 0.71–0.90) during an additional median 2.3 years unblinded follow-up. We note differences between the compound's mode of action, for example, dalcetrapib is a prodrug and modulates rather than competitively inhibits CETP activity and does not affect HDL2 and pre-β HDL subfractions[16]. While dalcetrapib likely produced suboptimal CETP inhibition, there is convincing evidence from animal models and human studies that torcetrapib exerts an off-target effect on the adrenal gland through aldosterone that may explain the observed blood pressure elevation, which in turn may have contributed to the increased risk of CVD[17]. It is interesting to

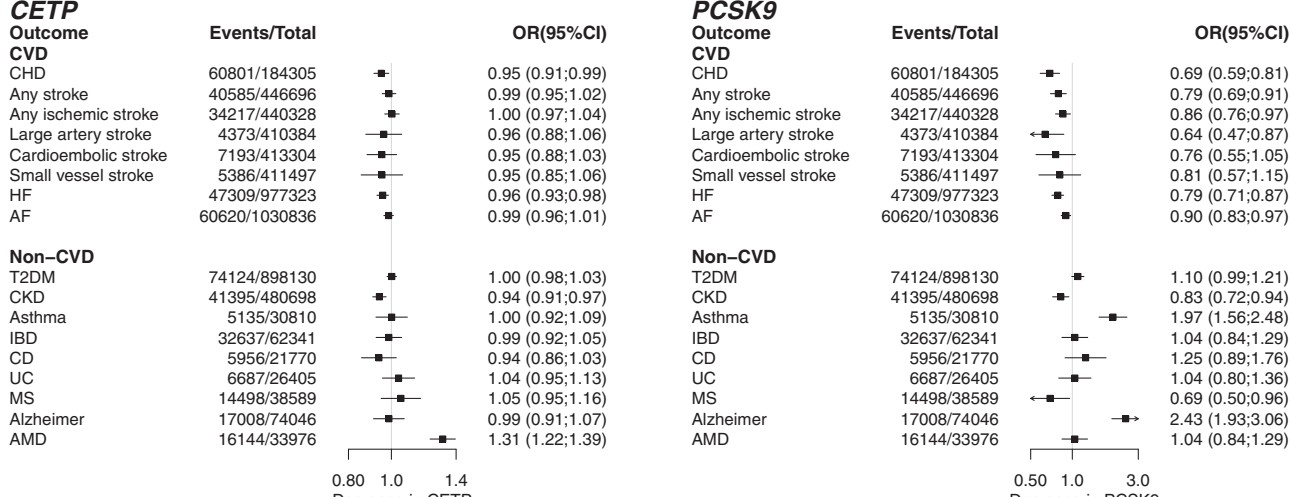

**Fig. 3 The drug target Mendelian randomization effects of lower CETP and PCSK9 concentration on clinical end-points.** N.B CHD: coronary heart disease, HF: heart failure, AF: atrial fibrillation, T2DM: type 2 diabetes mellitus, CKD: chronic kidney disease, IBD: inflammatory bowel disease, CD: Crohn's disease, UC: ulcerative colitis, MS: multiple sclerosis, AMD: age-related macular degeneration. Error bars reflect 95% confidence intervals, where the central dot represents the odds ratio. The number of events and total samples are provided in the figure as "Events/Total".

note that evacetrabip and anacetrapib both had significant SBP and DBP increasing effects (albeit smaller in magnitude compared to torcetrapib), while dalcetrapib showed an SBP decreasing effect which was in line with the CETP MR results. Taken together, the presented RCT and drug target MR findings suggest that CETP is a viable target and CETP inhibition a viable mechanism for CVD prevention and that the heterogeneous clinical effects of evaluated CETP inhibitors, e.g., the increased risk of mortality and CVD by torcetrapib or the modest lipids effects of dalcetrapib, are likely to be compound rather than target-related[16].

As well as enabling separation of on- vs. off-target effects of CETP inhibition, drug target MR analyses facilitate investigation of CETP effects beyond those investigated in clinical trials. Our analyses showed that lower CETP concentration was associated with a lower risk of CHD (OR 0.95 per μg/ml lower CETP concentration; 95% CI: 0.91–0.99), HF (OR 0.95; 95% CI: 0.92–0.99), and CKD (OR 0.94; 95% CI: 0.91–0.98), CKD (OR 0.94, 95% CI: 0.91–0.97) and a higher risk of AMD (OR 1.31, 95% CI: 1.22–1.40). The CETP analyses using genetic associations with LDL-C, HDL-C, or TG, as downstream proxies for lower CETP concentration or activity additionally suggest that CETP inhibition might protect against T2DM; consistent with findings from CETP-inhibitor trials[11]. Genetically-instrumented lower PCSK9 concentration was associated with a lower risk of CHD, HF, and CKD, and additionally with any stroke, ischemic stroke, AF, MS, as well as an increased risk of asthma and AD[18]. While the pQTL CETP analysis did not show a convincing effect on AD (Fig. 3), the lipid weighted analyses provided directionally discordant results, observing a risk increasing AD effect when weighting by LDL-C (Supplemental Fig. 3), compared to a protective effect CETP mediated by increased HDL-C using an MVMR model accounting for LDL-C.

We note that the "biomarker weighted" drug target MR results should not be confused with MR analyses designed to evaluate the causal relevance of major lipid fractions; utilizing genetic variants selected from throughout the genome[19]. In the presence of post-translational pleiotropy[19], where perturbation of a protein could affect multiple downstream biomarkers, some of which may lie on the causal pathway to disease and others not, biomarker-weighted drug target MRs, using *cis* instruments, do not necessarily provide evidence on the possible mediating pathway of the

drug target on disease[19] (unless employing MVMR) and instead reflect the effect of drug target perturbation regardless of the downstream pathways to involved in disease risk.

Our analyses are in line with the previous biomarker weighted analyses of CETP and CHD using Apo-B biomarker weights[20]. The current manuscript adds to these results by weighting the genetic effect by CETP itself, thereby more closely modeling the effect of CETP inhibition by a specific drug, and by considering 190 phenotypes including many non-CVD outcomes. We found that CETP and PCSK9 have distinct patterns of effect on apolipoprotein concentration as well as lipoprotein sub-fractions assayed through NMR spectroscopy. Lower CETP and PCSK9 concentration both decrease Apo-B, but CETP additionally increases HDL-C and Apo-A1 concentration, while also decreasing VLDL-C concentrations and to a lesser degree IDL-C concentrations. PCSK9 instead predominantly affects LDL-C sub-fractions, which are minimally affected by CETP.

As described above, the *cis*-MR analysis weighted by CETP concentration indicated a causal effect of lower CETP concentration on HDL-C, VLDL-C, IDL-C, as well as Apo-A1 and Apo-B, but only to a lesser extent with LDL-C measured by NMR spectroscopy using the Nightingale platform. This finding is consistent with the findings of Blauw et al in 5672 participants from the NEO study[21] and of Kettunen at al.[22], in Finnish cohorts and the INTERVAL study. We did however identify a strong *cis*-MR CETP effect on LDL-C when LDL-C was measured using clinical chemistry methods. Notably, by contrast, our MR analysis of the effect of PCKS9 on lipids and lipoproteins showed the expected association with LDL-C assayed both by non-size specific methods and by the Nightingale-NMR platform. A potential explanation for this discrepancy may be found in Tikkanen et al. who identified a strong correlation between clinical chemistry measured LDL-C and a derived "clinical" LDL-measure from the Nightingale NMR platform[23] which incorporates the additional lipoprotein subfractions VLDL-C, IDL-C, and lipoprotein(a), subfractions also included as part of the clinical chemistry-based assay methods for LDL-C. However, other NMR-based methods from Liposcience and Health Diagnostic Laboratory have shown limited agreement with different methods in the measurement of LDL-C[24].

We note that in the REVEAL trial of anacetrapib and the TULIP trial of obicetrapib, these specific, potent CETP inhibitors

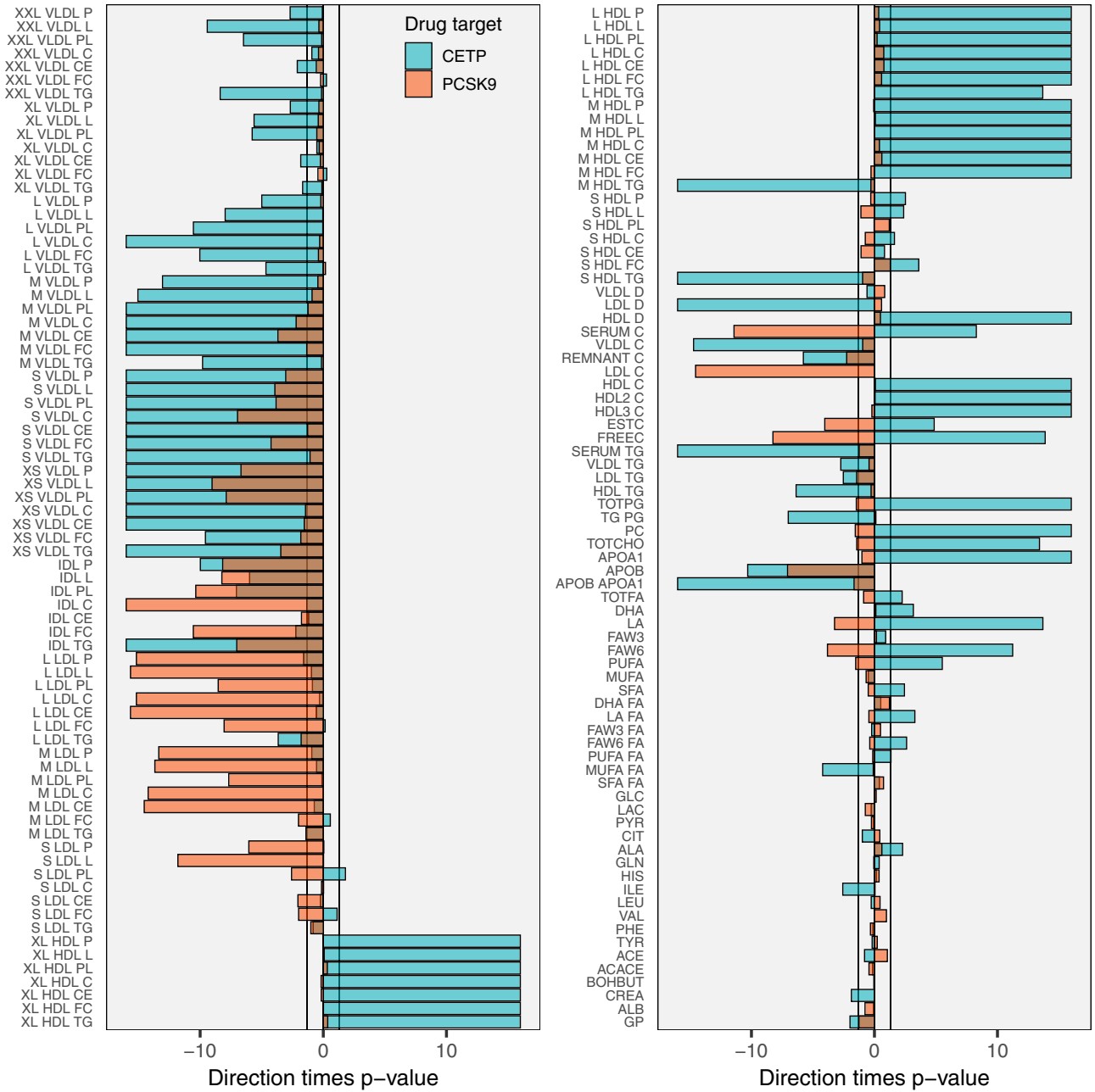

**Fig. 4 The drug target Mendelian randomization effects of lower CETP and PCSK9 concentration on NMR-measured metabolites.** N.B Results are provided as $-\log_{10}$(p-values) times effect direction, with the x-axis limits set to ±16. Bars are semi-transparent and plotted on top of each other to directly compare the two drug targets in their NMR measured lipids effect estimates. The vertical lines at ±1.3 represent the traditional p-value threshold of 0.05. p-values were calculated from two-sided Z-test statistics, without multiplicity correction. Analyses were based on a sample size of 33,029 subjects.

consistently show a LDL-C lowering effect (when LDL-C is measured using non-NMR based methods) and that the reduction in CHD events in the REVEAL trial of anacetrapib appears to be in proportion to its lowering of non-HDL-C (which is calculated as total cholesterol - HDL-C, and includes IDL-C and small-VLDL-C); where the relationship between non-HDL-C lowering and reduction in CHD event rate is set in the context of the prior trials of statins, as summarized by the Cholesterol Treatment Trialists collaboration (see Fig. 5S in ref. [11]). To further attempt to explain these discrepancies, it would be important to perform analyses using both size-specific (NMR) and non-size-specific LDL-C assay methods in the same trial participants.

We additionally set out to identify possible mediating lipid pathway between CETP and disease, performing multivariable MR (MVMR). These analyses suggest that the CHD effect of CETP was partially mediated by HDL-C; also noting a potential Apo-B mediation signal. We similarly identified HDL-C as a likely mediator of the CETP effects on T2DM and AMD. Compared to CETP, the MVMR analysis of PCSK9 convincingly showed LDL-C to be the most likely mediating lipid relative to HDL-C. Collectively, these findings suggest that, although sharing salutary effects on certain clinical endpoints, the mechanisms through which CETP and PCSK9 act are likely to be target-specific, as well as outcome-specific. Our findings support the

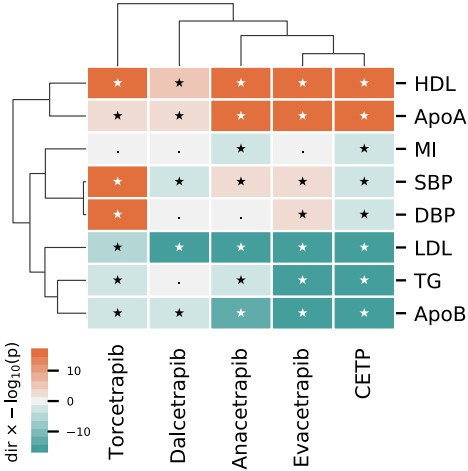

**Fig. 5 A cluster analysis comparing the on-target Mendelian randomization effect of lower CETP concentration to effects from CETP inhibiting compounds.** N.B CETP (MR estimates) and drug compound are ordered by columns, with specific outcomes listed in the rows. Effects are presented as −log10(p-values) × effect direction, where the CETP effect is orientated towards the CETP decreasing direction. p-values were calculated from two-sided Z-test statistics, without multiplicity correction. We note that p-values can be mapped to z-statistics, for example for a p-value of 0.05 we have −log₁₀(0.05) = 1.3, which can be mapped to a z-statistic of 1.96. Clustering was performed on the square root of the −log₁₀(p-values) × effect direction, with the p-value truncated to $10^{-60}$ to ensure enough difference between the CETP compound effect on changes in lipids. Associations with a p-value below 0.05 are indicated with a star. The dendrograms represent clustering by outcome (rows) and compound/drug target (columns). Point estimates (OR, MD), confidence intervals, and p-values are presented in Supplemental Tables 3, 6, and 7.

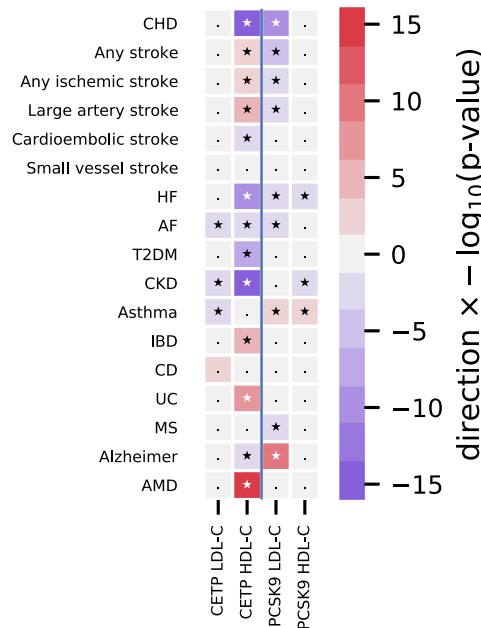

**Fig. 6 The multivariable drug target MR assessing the possible LDL-C and HDL-C mediating effects of CETP and PCSK9 concentration on the incidence of clinical events.** N.b. Results are colored by −log₁₀(p-values) × effect direction, with starred tiles indicating results with a p-value > 0.05. p-values were calculated from two-sided Z-test statistics, without multiplicity correction. Effects were orientated towards the canonical drug target effect direction: decreasing for LDL-C and increasing for HDL-C.

proposal that inhibiting both proteins jointly may elicit benefit, through (multiple) distinct lipid pathways.

Some previous drug target MR studies have attempted to quantify the anticipated effect of a drug targeting the same protein. For example, the anticipated effect of CETP inhibition on CHD risk is a reduction of 40% when weighted by one mmol/L lower LDL-C concentration (Supplemental Fig. 3). While of potential interest, there are some caveats that suggest that drug target MR analysis may be more useful as a reliable test of effect direction. This is because drugs that inhibit a target do so usually by modifying its function not its concentration, whereas genetic variants used in MR analysis usually affect protein expression and therefore concentration. However, for enzymes like CETP, measured activity reflects both the amount of available protein in circulation as well as activity per unit concentration. Thus, on both theoretical grounds and through empirical examples[20,25,26], MR analyses using variants in a gene encoding a drug target that affects its expression (or activity) have reproduced the effect direction of compounds with pharmacological action on the same protein[20,25,26]. Given the typically non-linear drug dose response, and the typically modest explained variance but relatively common genetic variants have on the level or function of a protein, it may be challenging to go beyond inferences on the direction and rank order of effects, to estimating the magnitude of a drug target effect, simply on the basis of genetic evidence. MR analyses assess the effect of target modulation in any tissue, whereas certain tissues may be inaccessible to a drug either because of its chemistry or physiological barriers. Furthermore, RCTs are closely monitored, and followed for a fixed period, allowing for exploration of induction-times[11]. MR estimates are considered to

reflect a life-long exposure, but in the absence of serial assessment, possible changes across age are difficult to explore, as are disease induction times. For these reasons, we suggest that drug target MR offers a robust indication of effect direction but may not directly anticipate the effect magnitude of pharmacologically action of a drug on its target, even in the absence of off-target drug compound effects. Findings such as the observed increased risk of AMD (from lower CETP), or of asthma and Alzheimer's disease (from lower PCSK9), or the apparent protective effect on MS (from lower PCSK9) provide inference on the likely consequences of protein inhibition. However, whether pharmaceutical compounds targeting these proteins elicit similar effects depends on both the duration of drug exposure, as well as the potential for a drug to access the relevant tissues. For example, monoclonal antibody PCSK9 inhibitors may not cross the blood brain barrier. Nevertheless, these findings are relevant for pharmaceutical companies, as well as medicines regulators undertaking post-marketing surveillance of agents targeting these proteins.

In conclusion, previous failures of CETP inhibitors are likely related to suboptimal target inhibition (dalcetrapib), off-target effects (torcetrapib), or insufficiently long follow-up (evacetrapib). The present drug target MR analysis, consistent with findings from the anacetrapib trials, anticipates that on-target CETP inhibition decreases CVD risk. MR analyses additionally suggest a reduction in the risk of type 2 diabetes and kidney disease, but an increased risk of age-related macular degeneration.

## Methods

**Systematic review and meta-analyses of CETP inhibitor effects.** CETP inhibitor trials with at least 24 weeks of follow-up (irrespective of phase) were identified through a systematic review using a pre-specified search strategy (Supplementary note 1) of MEDLINE and OVID, supplemented by clinicaltrials.gov. Parallel-group

RCTs were included regardless of the comparator (placebo or active therapy). Due to the cessation of randomization and potential for contamination by unblinding participants we excluded any post-trial follow-up data. Treatment effects were extracted (by NH and AFS) on lipids, lipoproteins, blood pressure, the incidence of ACM and cardiovascular endpoints: any CVD (defined as CV death, MI, any stroke, and angina hospitalization), fatal CVD, any MI (including CHD), fatal MI, any stroke (ST; including ischemic, hemorrhagic and other strokes), ischemic stroke, hemorrhagic stroke, and HF. Treatment effects on continuous traits (mean differences (MD)) were extracted as the between-group difference in percentage change from baseline[27]. Additional data were extracted on compound dose and potency, trial participants, and setting. Compound-specific clinical trial data were meta-analyzed using the inverse-variance weighted method applying both fixed and random effects. The Q-statistic[28,29] was used to test for the presence of between compound heterogeneity.

We furthermore, explored small study effects using funnel plots, and used meta-regression to (indirectly) explored treatment effect modification. Specifically, meta-regression attempted to associate trial-specific characteristics to CVD treatment effects. Here we focussed on mean age, mean BMI, the proportion of women, treatment effects on LDL-C, Apo-B, and HDL-C (all three as the difference in percentage change from baseline). We note that regressing the compound CVD effect on the compound biomarker effect(s) constitutes an instrumental variable analysis[30], as well as an assessment of treatment effect modification.

**Mendelian randomization analysis.** Drug target MR analysis[19] utilizes (cis)-variants in, or near, a drug target encoding gene to obtain a causal estimate of the protein effect on multiple outcomes. Genetic associations with an outcome (e.g., CHD) are regressed on genetic associations with the drug target protein concentration or, alternatively, with biomarkers distal to the protein. Under the assumption that all the effects of the genetic variants on an outcome are mediated by the drug target protein (absence of pre-translational horizontal pleiotropy), the slope represents an estimate of the drug target effect. Here we used genetic effect estimates on the concentration of the encoded protein (CETP or PCSK9) as the primary exposure of interest, repeating the analyses using genetic effect association with LDL-C (for CETP and PCSK9), HDL-C (for CETP), and TG (for CETP), representing biomarkers known to be affected by the corresponding protein (available from the GLGC[31] consortium).

We additionally employed MVMR to explore potential mediating pathways of protein (CETP or PCSK9) effects. MVMR evaluates mediation through joint modeling of multiple candidate mediators[32], generalizing traditional "univariable" MR in much the same way as linear regression can be extended to consider multiple (multivariable) risk factors. Due to our focus on the small cis region encoding either PCSK9 or CETP, the number of available variants was limited, diminishing the number of potential mediators we could jointly consider. Hence, we concentrated on joint modelling of (1) HDL-C and LDL-C and (2) HDL-C and Apo-B. Both drug target MR and MVMR invoke a no-horizontal pleiotropy assumption. While a drug target MR will be unbiased in the absence of any pre-translation pleiotropy[19], MVMR requires a stronger no-horizontal pleiotropy assumption, additionally assuming all mediators of the protein effect on disease are included in the model.

To reduce the risk of "weak-instrument bias"[33], we selected genetic variants with an F-statistic of 15 or higher (Supplemental Tables 9–14). We used a two-staged MR-paradigm, where genetic associations with the exposure and outcome were derived in independent samples, ensuring that any remaining weak-instrument bias attenuates towards the null (conservative estimates)[33]. Given the differences in coverage between the various outcome GWAS, variants were clumped to an R-squared of 0.40 *after* linking the exposure variants to a specific outcome GWAS (maximizing precision). Residual linkage disequilibrium (LD) was modeled using a generalized least squares[34,35] inverse-variance weighted-estimator, and an external correlation structure (random 5000 UK biobank, (UKB) sample). The possibility of bias due to horizontal pleiotropy was minimized by focussing on a cis genetic region, excluding variants with large leverage or outlier statistics[19,36], and using the Q-statistic to identify remaining violations.

Findings from the cis-MR analysis of CETP were compared to effects observed in trials (for outcomes shared by the trial and MR analyses) using hierarchical clustering.

**Selection of genetic instruments.** Genetic associations with CETP concentration (protein quantitative trait loci; pQTLs) were extracted from a GWAS on circulating CETP concentration[14]. Genetic variants were selected based on residency within a narrow window around *CETP* (Chr 16: bp: 56,961,923–56,985,845; GRCh38)[14]. For the PCSK9 drug target MR, we selected variants associated with PCSK9 concentration[37] using the following window: 55,037,447–55,066,852 bp (Chr 1; GRCh38). Variants with a minor allele frequency below 0.01 were removed.

Results were presented as MD or OR with a 95% confidence interval (95% CI) coded toward the canonical drug target effect direction; i.e., toward lower circulating protein, LDL-C, and TG concentration, and a higher HDL-C concentration. CETP concentration was reported as $\mu$g/ml while PCSK9 concentration was reported as log-transformed ng/ml.

**Reporting summary.** Further information on research design is available in the Nature Research Reporting Summary linked to this article.

## Data availability

The processed MR and trial data generated in this study have been deposited in the UCL Research Data Repository database under accession code 13686247.v3. The supplemental tables include all the aggregated data (effect estimates, standard errors, and so on) presented here. All source GWAS data are publicly available (see URL supplied in Supplementary Table 15) including 60,801 CHD cases from CardiogramplusC4D[38] (http://www.cardiogramplusc4d.org/); 40,585 stroke cases (subtypes) from MEGASTROKE[39] (http://www.megastroke.org/index.html); 47,309 HF cases from HERMES[40] (https://www.ebi.ac.uk/gwas/publications/31919418), 60,620 atrial fibrillation cases from AFgen[41] (http://csg.sph.umich.edu/willer/public/afib2018/), 17,008 Alzheimer's disease[42] cases(https://www.niagads.org/), 16,144 age-related macular degeneration events from IAMDGC[43,44] (http://amdgenetics.org/), and genetic associations with NMR measured circulating lipoprotein subfractions and other metabolites were available from a meta-analysis of Kettunen et al.[45], and UCLEB[46] (n: 33,029, http://www.computationalmedicine.fi/data/NMR_GWAS/). Additionally, the following resources were sourced: major circulating lipid sub-fractions or apolipoproteins (LDL-C, HDL-C, TG, lipoprotein A [Lp(a)], apolipoprotein B, apolipoprotein A1), pulse rate, glucose and HbA1c, leukocytes, lymphocytes, monocytes, neutrophil counts, and C-reactive protein, using data from the UK Biobank (UKB - http://www.nealelab.is/uk-biobank). Blood pressure (systolic and diastolic) data were available from Evangelou et al.[47] (https://grasp.nhlbi.nih.gov/FullResults.aspx). Carotid artery intima-media thickness was available from a meta-analysis of the Cohorts for Aging Research in Genomic Epidemiology (CHARGE)[48] and University College London Edinburgh Bristol (UCLEB)[46] (https://www.ncbi.nlm.nih.gov/projects/gap/cgi-bin/study.cgi?study_id=phs000930.v6.p1). The CKDGen consortium provided GWAS associations on blood urea nitrogen (BUN), estimated glomerular filtration rate (eGFR), and chronic kidney disease[49] (http://ckdgen.imbi.uni-freiburg.de/). Bone mineral density[50] GWAS data were obtained from GEFOS Consortium (http://www.gefos.org/). Genetic associations with "general cognitive function" were obtained from a meta-analysis of CHARGE, COGENT, and UKB[51] (https://www.thessgac.org/data). Data were extracted on type 2 diabetes[52] from DIAGRAM (http://diagram-consortium.org/index.html); asthma[53] (http://ftp.ebi.ac.uk/pub/databases/gwas/summary_statistics/GCST006001-GCST007000/GCST006911/), inflammatory bowel disease[54], Crohn's disease[55] and ulcerative colitis[56] from IIBDGC (https://www.ibdgenetics.org/); multiple sclerosis[57] from the IMSG consortium (https://imsgc.net/). Finally, genetic association with CETP or PCSK9 concentration was sourced from Blauw et al.[14] (https://www.ahajournals.org/doi/full/10.1161/CIRCGEN.117.002034) and Plot et al.[37] (https://pubmed.ncbi.nlm.nih.gov/29748315/).

## Code availability

Analyses were conducted using Python v3.7.4 (for GNU Linux), Pandas v0.25, Numpy v1.15, Seaborn v0.11.5, R v4.0. 3 (for GNU Linux), ggplot2 v3.3.5, metafor[58] v3.0.2 and forestplot v1.10.1. The Python and R scripts and data necessary to generate the illustrations have been deposited through the UCL Research Data Repository: https://doi.org/10.5522/04/13686247.v3.

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

## Acknowledgements

This research has been conducted using the UK Biobank Resource under Application Number 12113. The authors are grateful to UK Biobank participants. We gratefully acknowledge the support of UCLEB and CHARGE. Funding and role of funding sources: A.F.S. is supported by BHF grant PG/18/5033837 and the UCL BHF Research Accelerator AA/18/6/34223. C.F. and A.F.S. received additional support from the National Institute for Health Research University College London Hospitals Biomedical Research Centre. M.G.M. is supported by a BHF Fellowship FS/17/70/33482. A.D.H. is an NIHR Senior Investigator. This work was supported by the UKRI/NIHR Strategic Priorities Award in Multimorbidity Research (MR/V033867/1). This work was additionally supported by a grant [R01 LM010098] from the National Institutes of Health (USA). We further acknowledge support from the Rosetrees and Stoneygate Trust. The UCLEB Consortium is supported by a British Heart Foundation Program Grant (RG/10/12/28456). T.R.G. receives support from the UK Medical Research Council (MC_UU_00011/4). D.O.M.K. is supported by the Dutch Science Organization (ZonMW-VENI Grant 916.14.023). A D.H. receives support from the UK Medical Research (MC_UU_12019/1). M.K. is supported by the Wellcome Trust (221854/Z/20/Z), the UK Medical Research Council (MR/S011676/1, MR/R024227/1), National Institute on Aging (NIH), US (R01AG062553), and the Academy of Finland (311492). D.A.L. is supported by a Bristol BHF Accelerator Award (AA/18/7/34219) and BHF

Chair (CH/F/20/90003) and works in a unit that receives support from the University of Bristol and the UK Medical Research Council (MC_UU_00011/6). D.A.L. is a National Institute of Health Research Senior Investigator (NF-0616-10102). N.F. is supported by the National Institutes of Health (R01-MD012765, R01-DK117445, R21- HL140385). P.C. is supported by the Thailand Research Fund (MRG6280088. UK Biobank was established by the Wellcome Trust medical charity, Medical Research Council, Department of Health, Scottish Government, and the Northwest Regional Development Agency. It has also had funding from the Welsh Assembly Government and the British Heart Foundation. Infrastructure for the CHARGE Consortium is supported in part by the National Heart, Lung, and Blood Institute grant R01HL105756. The preliminary meta-analysis of RCT data were presented at BPS 2018 by NH. The preprint version of this paper has been deposited on medrxiv: https://doi.org/10.1101/2020.09.07.20189571.

## Author contributions

A.F.S., A.D.H., C.F. contributed to the idea and design of the study. A.F.S. and N.H. performed the systematic review and meta-analysis. A.F.S. and C.F. performed the drug target MR analyses. A.F.S. drafted the paper, and N.B.H., M.G.M., P.C., F.D., M.K., D.A.L., C.G., O.P., N.C., J.C.B., C.J.O., G.W., A.W., J.F.P., A.D.H., T.R.G., N.F., D.O.M.K., M.Z., R.S., A.D.H., and C.F. provided critical input on the analyses and the drafted paper. A.F.S. performed the here presented analyses, had full access to all the data in the study, and takes responsibility for the integrity of the data and the accuracy of the data analysis.

## Competing interests

A.F.S. has received Servier funding for unrelated work. M.Z. conducted this research as an employee of BenevolentAI. Since completing the work M.Z. is now a full-time employee of GlaxoSmithKline. Since completing the work C.J.O. is now a full-time employee of Novartis Institutes for Biomedical Research. D.A.L. has received support from Roche Diagnostics and Medtronic Ltd. for research unrelated to this paper. T.R.G. receives funding from GlaxoSmithKline and Biogen.
