## [Peer Review File · Nature Communications]

REVIEWER COMMENTS

Reviewer #1 (Remarks to the Author):

Dear authors ,

I have read with interest your impressive work on Mendelian randomization of the CETP gene as well your meta-analysis of CETP-inhibitor trials.....I have a number of issues that merit consideration and that I would like to share with youbeing a clinician I would like to start with the meta-analysis first

CETP inhibitor trials

First of all , the only two CETP-inhibitors that are still in clinical development are dalcetrapib in the Dal-Cor CVOT that reports out in 2021 and dalcetrapib which will go into phase III development in 2021 as wellATH03 , DRL-17822 , JTT-302 , K-312 as well as rocacetrapib are all discontinued in their clinical development and are no longer worth mentioning in Table S1.

In addition , the word meta-analysis sounds impressive and suggests a final conclusion , but unfortunately the sum is only as good as its componentsthe only studies I would include in this meta-analysis are ACCELERATE , Dal-Outcomes , and REVEAL and then in case of the latter the 6.4 year follow up presented at the AHA meetings in 2019 and not the 4 year follow-up.....why not include the rest ?

Torcetrapib by virtue of its molecular structure can penetrate the zona glomerulosa of the adrenal and elicits the production of aldosterone and cortisol with ensuing hypertension and electrolyte changes..in addition it penetrates the endothelial lining of medium to large arteries and stimulates the production of endothelin-I leading to vasoconstriction in arterial beds

All these effects occur in humans , but also in rodent models that lack CETP , so this is clearly an off-target effect , that was not shown in any other CETPi and completely confuses a meta-analysis that looks at MACE and other endpoints...so , in my opinion torcetrapib data can only be used to study effects on lipids and lipoproteins and not endpoints of any kind.....

Second , the REVEAL and ACCELERATE studies show that the Kaplan-Meier curves only diverge at the 2 year mark , especially when the CETP-I is used on top of high dose statins...so , including all these small studies with short follow-up is not helpful at all and only introduces bias towards the null-hypothesisIn fact , these three trials combined have included over 50.000 patients , so more than enough power to make strong conclusions and absolutely no need to revert to smaller studies with shorter follow-up...

I am actually surprised why the findings from the 2017 JAMA paper by Ference et al is not given more prominence in your paper.....that manuscript once and for all showed MR evidence for low CETP haplotypes to lead to lower CHD risk , but maybe more importantly showed for anacetrapib , evacetrapib as well as obicetrapib that CETPi as monotherapy reduce apoB by double the percentage

as CETPi used on high doses of potent statins....data from the JAMA 2011 , Lancet 2015 and NEJM 2017 papers

But maybe , one of the biggest problems with this meta-analysis is the supposed efficacy of the different drugs on LDL-C reductions ; It is listed as -38% for anacetrapib , -37% for evacetrapib etc etc ...These numbers are clearly wrongas Michael Holmes clearly shows in Nature Reviews Cardiology you have to use betaquant to reliably measure LDL lowering with a CETPi and in fact the -41% LDL-C reduction in REVEAL , as assessed by Friedewald or the direct enzymatic assay , is only -17%

One can immediately appreciate that this has to be right when the apoB reduction is considered ; also -18% and completely in line with the LDL-C reduction as assessed by betaquant....

This reviewer has been in touch with Eli Lilly since the LDL-C reductions as reported for ACCELERATE in the NEJM are clearly wrongthe overall apoB reduction in that study is around -19% placebo controlled , but only -15% from baseline whereas the reported LDL-C reduction is -31 % ...in the M&M it is stated that LDL-C was measured by a direct assay and by betaquant , but the betaquant results are NOT in the NEJM paper , those are the direct assay resultsthe real results will be around 20 % and not morethis is also true for torcetrapib by the way , but is not relevant for dalcetrapibso , all your calculations have to be redone since the numbers you list are a complete over estimation of the real LDL-C lowering effect....only betaquant results are reliable and if not available apoB reductions are the next best.....

In fact , one can even question if dalcetrapib should be part of this analysis at all....in elegant crystallography studies Shenping Liu showed in the J Biol Chem in 2012 that dalcetrapib has a very different chemical structure than any other CETPi , binds to a different pocket on the CETP-protein , does not block the entrance to the tunnel that transports cholesterylesters and binds CETP covalently to the HDL particlein fact , JTT-705 or dalcetrapib came out of a large scale screen for ACAT inhibitors and was so different to the rest that Roche scientists introduced the term CETP-modulator instead of inhibitor.....

It has no effect on apoB , so it will have no effect on MACEplease consider the clinical update in JAMA 2020 on Lipids and Lipoproteins by Ference et al that conclusively sums up the evidence that apoB is a better risk marker and a better intermediate marker for risk reduction than LDL-C ...I would find it appropriate to use apoB instead of LDL-C for your paper since for apoB there can be no misunderstanding as to how to measure it.....please realize all data for torcetrapib are Friedewald based and most data for anacetrapib as well as evacetrapib are direct enzymatic assay based...so , the majority of LDL data in your analysis are not correct

As a last remark , this reviewer sees no relevance at all for lipoprotein subfractions.....no information beyond apoB has shown to be relevant.....

Mendelian randomization

It is quite evident that the apoB lowering effect of CETP-inhibitors must be driven by LDL receptor uptake since CETPi work fine in heFH but not at all in hoFHso it is intuitive why CHD in your MR assay works through LDL-C lowering...what is counterintuitive is the fact that TG lowering (which is irrelevant with CETPi) and HDL-C increase , which was shown in REVEAL to be irrelevant but safe for

CHD also lights up in your MR plotsyou need to devote much more discussion to this because here clearly trials are at odds with MR data

What is also incompatible with the evidence is that LDL-C decrease would lower type II DM with CETPiin fact this beneficial effect on HbA1C , HOMA-IR and the incidence of new onset type II is seen with all CETPi's and the best data set is with dalcetrapib , very recently published by Greg Schwartz in Diabetes Care that showed an incredible 26% lower incidence of type II at the end of Dal-Outcomes....dalcetrapib has NO effect on apoB containing lipoproteins , so your analysis cannot be truethe only thing these 4 compounds have in common is the HDL increase and the higher cholesterol efflux capacity....

In fact there is a very sound hypothesis for this mechanism ; removal of toxic sterols from the beta cells in the pancreas by pre-beta HDL , which production is enhanced by at least 30% by CETPi as shown in TULIP in Lancet in 2015 by Hovingh et al

This brings me to my last remark ; looking at your HDL plot , the point estimate for protection against type II DM overlaps with the point estimate for protection Alzheimers diseasethere is a revival going on for the sterol hypothesis in AD , see van der Kant et al in Cell Stem Cell , 2019

Quite intriguingly this is based on the lack of efficient cholesterol efflux in apoE4/E4 carriers.....CETPi increase plasma apoE by about 30 to 50% and this apolipoprotein is small enough to cross the BBB at the choroid plexus , so this might be the link between low CETP and protection against Alzheimers which was shown first in centenarian Ashkenazim in New York in JAMA 2015.....

I wonder why you choose to discuss AD for PCSK9 , but not for CETPi ? The biological hypothesis is much stronger for the first than for the latter

Reviewer #4 (Remarks to the Author):

Schmidt et al report an interesting meta-analysis on cholesteryl ester transfer protein genes/levels/inhibitors focusing on their association with surrogate cardiovascular and clinical outcomes, and potentially informing on future drug development and adoption.

Despite the work strengths, I recommend addressing the following comments:

1. Methods and Results: Test between-agent differences with network or multivariate meta-analysis.
2. Methods and Results: Appraise with meta-regression if some patient features (eg other conditions, ancillary therapy) impact on the risk-benefit balance of CETP inhibitors.
3. Methods and Results: Appraise with meta-regression or multivariate meta-analysis which surrogate effects (eg on LDL, CRP or TG) are most predictive of clinical benefits.

4. Methods and Results: Can you identify the optimal dosage to achieve a given effect on surrogate endpoints or on hard clinical outcomes?

5. Methods and Results: Perform detailed small study effect analyses (eg funnel plots, regression tests).

6. Discussion: Elaborate on the most appropriate placement of CETP inhibitors in the context of other lipid lowering therapies, and also on the appropriate per-patient cost of eventually approved agents.

Reviewer #5 (Remarks to the Author):

This is a very clear and thorough study comparing the effects of HDL-raising therapies with genetic variants that, in part, mimic the effects of those therapies. The results are interesting and provide evidence of potential efficacy and adverse events. The genetic evidence complements and goes beyond the trial evidence because many more conditions can be examined in large existing datasets.

1. The most important comment – i cannot see that the authors have tried to distinguish the HDL-C raising effects from the LDL-C , triglyceride etc lowering effects using the CETP genetic instruments. Weighting the exposure by the main effects on LDL-C, triglycerides and HDL-C etc does not answer the question of what are the marginal effects of these different risk factors because it still focuses on main effects that likely include effects on several pathways (e.g. genetically instrumented lower CEPT alters LDL-C by about a 1/3rd of the effect it raises HDL-C). The authors briefly mention this in the discussion but this is another advantage of the genetic approach and a key question after the on vs off target Q. Is the CETP benefit on CVD outcomes on target and through HDL-C or on-target and through LDL-C, Lipoprotein A , triglycerides etc independent of the other factors – especially independent of LDL-C the strongest risk factor for CVD (although IpA looks to be the most strongly influenced by the genetics of CETP ?). This question can be answered through multi-variable MR. Have the authors considered a multivariable MR ?

Also additional main points:

2. Figure 5 is a key plot but is not very informative as presented. Why not present the effects and their 95% CIs as a forest plot – the MR estimates scaled to a 1 unit / 1SD increase in the exposure and alongside the actual drug effect ? shades representing p values is not very transparent as to how the effects actually compare. I am not even sure which outcomes each cell refers to.

3. I accept that there are caveats and differences between the short term and specific effects of drug action compared with long acting more widespread perhaps effects of genetic variants, but do the authors want to expand on the need to monitor patients on PCSK9 inhibitors for signs of dementia more closely ? this genetic finding , whilst not the primary hypothesis, suggests such a measure might need more follow up work ?

4. Clumping by r^2 of 0.4 sounds too permissive, in that two variants with an r^2 of 0.39 are not providing independent information – indeed they will explain 0.39 of the variance in each other. It is not obvious from the main methods or results how many variants were used for each genetic instrument.

Minor points

1. Why exclude variants < 0.05 for one gene, but not for another ?
2. In lines 241-245, describing the CETP MR results on major endpoints, it would help to add “lower” and “higher” to the text to help the reader. Likewise abstract lines 70-74
3. Line 235 – do you mean Lower “genetically instrumented” CETP concentrations ?
4. Fig S2 – why is there no data for CETP MR weighted by LDL-C for AMD ?

Tim Frayling, Exeter

Reviewer #1

Dear authors ,

I have read with interest your impressive work on Mendelian randomization of the CETP gene as well your meta-analysis of CETP-inhibitor trials.....I have a number of issues that merit consideration and that I would like to share with youbeing a clinician I would like to start with the meta-analysis first
CETP inhibitor trials

1) First of all , the only two CETP-inhibitors that are still in clinical development are dalcetrapib in the Dal-Cor CVOT that reports out in 2021 and dalcetrapib which will go into phase III development in 2021 as wellATH03 , DRL-17822 , JTT-302 , K-312 as well as rocacetrapib are all discontinued in their clinical development and are no longer worth mentioning in Table S1.

Response: Thank you. We have added a column to Table S1 including information on the development status of each of the compounds. We retain reference to discontinued compounds since the stated aim of the paper is to understand why the cholesteryl transfer protein inhibitor class of drug has yet to deliver a successful therapy. This requires evaluation of the progress of all compounds in this class.

2) In addition , the word meta-analysis sounds impressive and suggests a final conclusion , but unfortunately the sum is only as good as its componentsthe only studies I would include in this meta-analysis are ACCELERATE , Dal-Outcomes , and REVEAL and then in case of the latter the 6.4 year follow up presented at the AHA meetings in 2019 and not the 4 year follow-up.....why not include the rest ?

Response: Thank you for this comment. A meta-analysis is usually a culmination of a systematic review, which we undertook based on pre-specified inclusion and exclusion criteria. Omitting studies that fulfil the inclusion criteria and including studies that fail to meet these criteria violates the systematic review protocol. Regarding the inclusion of all RCTs with at least a 24 -week follow-up, we simply follow the Cochrane guidance on systematic review and meta-analyses of randomized trials. The report of the REVEAL trial mentioned by the reviewer refers to the extended, unblinded follow-up of trial participants (https://www.revealtrial.org/REVEAL_AHA_PTFU_Slides_2019_11_12.pdf), during which time the study drug was not administered. As such, this iteration of the REVEAL study findings fails to meet the pre-specified criteria for inclusion.

To further explain the rationale for the inclusion of studies we make the following amendments.

Page 16

‘Due to the cessation of randomization and potential for contamination by unblinding participants we excluded any post-trial follow-up data. ‘

Page 11:

University College London, Gower Street, London WC1E 6BT
Tel: 0044 (0)20 3549 5625
amand.schmidt@ucl.ac.uk
www.ucl.ac.uk

“The reduction in cardiovascular events seen in the REVEAL trial of anacetrapib (median follow-up 1,497 days; Supplementary Table 2) is consistent with the drug target MR results presented here. The manufacturer, Merck, did not seek marketing authorization for this drug, citing anticipated lack of regulatory support¹⁵. The evacetrapib ACCELERATE trial was terminated for futility after a median follow-up of 791 days, a time point before the benefits of anacetrapib emerged in the REVEAL trial (see Figure 1 of ref¹¹). We further note heterogeneity between compounds. For example, dalcetrapib is a prodrug and modulates rather than competitively inhibits CETP activity and does not affect HDL2 and pre-β HDL subfractions¹⁶. While dalcetrapib likely produced suboptimal CETP inhibition, there is convincing evidence from animal models and human studies that torcetrapib exerts an off-target effect on the adrenal gland through aldosterone that leads to blood pressure elevation, which may have contributed to the increased risk of CVD¹⁷. Nevertheless, it is interesting to note that evacetrapib and anacetrapib both had significant SBP and DBP increasing effects (albeit smaller in magnitude compared to torcetrapib), while dalcetrapib showed a SBP decreasing effect which was in line with the CETP MR results. Taken together, the presented RCT and drug target MR findings, suggest that CETP is a mechanistically viable target for CVD prevention, and the heterogeneous clinical effects of evaluated CETP inhibitors, e.g. the increased risk of mortality and CVD by torcetrapib or the modest lipids effects of dalcetrapib, are likely to be compound - rather than target-related¹⁶. “

3) Torcetrapib by virtue of its molecular structure can penetrate the zona glomerulosa of the adrenal and elicits the production of aldosterone and cortisol with ensuing hypertension and electrolyte changes..in addition it penetrates the endothelial lining of medium to large arteries and stimulates the production of endothelin-I leading to vasoconstriction in arterial beds

All these effects occur in humans , but also in rodent models that lack CETP , so this is clearly an off-target effect , that was not shown in any other CETPi and completely confuses a meta-analysis that looks at MACE and other endpoints...so , in my opinion torcetrapib data can only be used to study effects on lipids and lipoproteins and not endpoints of any kind.....

Response: The purpose of our study was precisely to understand which, if any, of the CETP-inhibitors evaluated in trials has off-target effects, and whether there are differences between compounds. The former was achieved by comparison of the effects of the different compounds observed in RCTs with those observed in the Mendelian randomisation analysis; and the latter by formally evaluating heterogeneity in treatment effects across the different compounds evaluated in trials. Our analysis supports the reviewer’s point that the BP elevating effect of torcetrapib is off-target, and that this BP-raising action might have contributed to the increased rate of cardiovascular events in participants randomised to torcetrapib rather than placebo in the phase 3 trial, since the opposite effect was observed in the genetic analysis. Our ability to make this statement is contingent on the inclusion of the trials of torcetrapib and indeed all the other compounds, for all relevant end points. Thus, the inclusion of the torcetrapib data serves to emphasise rather than downplay the point made by the reviewer.

To ensure that the aims of the study are completely clear, we have amended the final paragraph of the introduction on page 6 as follows:

“To address these uncertainties, we performed a drug target MR study of CETP, focusing on variants within the encoding gene (acting in cis) that are associated with circulating CETP concentration, to directly model the effects of pharmacological action on this target by a clean drug with no off-target actions. To evaluate potentially diverse effects of drug target perturbation, we combined drug target MR with a phenome-wide scan of over 190 disease biomarkers or clinical end-points relevant to cardiovascular as well as non-cardiovascular outcomes. We compared drug target MR effect estimates to compound-specific effect estimates derived from a systematic review and meta-analysis of CETP inhibitor RCTs. Assuming the developed CETP inhibitors sufficiently engaged the drug target, on-target failures would result in consistent treatment effects across all compounds, which should be similar to the on-target effect modelled through MR. Finally, drug target MR analyses of CETP and PCSK9, an archetypal LDL-C lowering drug target, were compared on their effects profile.

Here we show that previous failures of CETP-inhibitor drugs are likely compound rather than target related. Through drug target MR we show that on-target CETP inhibition is predicted to reduce the risk of several cardiovascular end points and diabetes but potentially increase the risk of age-related macular disease (AMD). Comparisons with PCSK9 drug target MR suggests that inhibiting both targets might provide additive benefit. “

4) Second , the REVEAL and ACCELERATE studies show that the Kaplan-Meier curves only diverge at the 2 year mark , especially when the CETP-I is used on top of high dose statins...so , including all these small studies with short follow-up is not helpful at all and only introduces bias towards the null-hypothesisIn fact , these three trials combined have included over 50.000 patients , so more than enough power to make strong conclusions and absolutely no need to revert to smaller studies with shorter follow-up...

Response: We thank the reviewer for this important comment. Inclusion of trials of at least 24 weeks duration but of shorter overall follow-up than phase 3 trials, informs the analysis of the effects of CETP inhibitor treatment on continuous risk factors such as blood lipid concentrations and blood pressure. The effects on clinical outcomes in such short-term studies may be distinct from longer follow-up studies given that patients may not have taken the drug for a sufficiently long period to provide benefit. The reviewer argues their inclusion in the meta-analysis of clinical end-points might introduce a bias towards the null. However, such trials are typically much smaller than phase 3 end-point trials, provide a very limited number of events, and therefore their contribution to the overall meta-analysis estimate is likely minor.

To illustrate, our analysis of anacetrapib included six trials, three of which provided data for CVD, whereupon the REVEAL trial contributed most samples. To explore the potential for small study induced heterogeneity, pulling effect estimate towards a null effect we conducted several sensitivity analyses.

First, we compared findings from the REVEAL trial alone with the meta-analysis results including all studies, and with a separate meta-analysis excluding REVEAL but including all remaining trials. The treatment effect estimates for CVD are: OR: 0.93 (95%CI 0.87; 1.00) for REVEAL only, 0.80 (95%CI 0.43; 1.48) for non-REVEAL studies, and 0.93 (95% 0.87; 1.00) for all trials of anacetrapib. As such, the inclusion of trials of anacetrapib other than REVEAL did not distort the findings toward the null.

Next, inspired by suggestions made by reviewer 4, we conducted a formal Egger test for small study heterogeneity and included funnel plots, again confirming the inclusion of smaller short-term studies did not impact our results:

Figure S2: Funnel plots assessing small sample treatment heterogeneity in cardiovascular (CVD) outcomes of CETP-inhibitors with more than one included trial (excluding the single study on Evacetrapib). None of the Egger tests for small study effects were significant at an alpha of 0.05.

We also note that our systematic review for evacetrapib trials only included the ACCELERATE study (as it was the only study that met our inclusion criteria), hence this is not an issue for the analysis of evacetrapib.

To recognise and discuss this important issue raised by the reviewer, we have included the following text on Page 8:

“Small study heterogeneity was explored using funnel plots (Supplemental Figure 2), which did not provide convincing evidence of differential CVD treatment effects by study size; although the number of available studies was limited. Given that the REVEAL trial of anacetrapib only showed treatment benefit after the first two years of follow-up¹¹, we analysed short follow-up studies separately: OR 0.80 (95%CI 0.43; 1.48) for CVD, and

compared this to REVEAL study CVD estimate: OR: 0.93 (95%CI 0.87; 1.00), showing no significant difference. "

5) I am actually surprised why the findings from the 2017 JAMA paper by Ference et al is not given more prominence in your paper.....that manuscript once and for all showed MR evidence for low CETP haplotypes to lead to lower CHD risk , but maybe more importantly showed for anacetrapib , evacetrapib as well as obicetrapib that CETPi as monotherapy reduce apoB by double the percentage as CETPi used on high doses of potent statins....data from the JAMA 2011 , Lancet 2015 and NEJM 2017 papers

Response: We thank the reviewer for drawing attention to the important 2017 JAMA paper from Ference and colleagues entitled 'Association of Genetic Variants Related to CETP Inhibitors and Statins with Lipoprotein Levels and Cardiovascular Risk' JAMA. 2017;318(10):947-956. doi:10.1001/jama.2017.11467 Published online August 28, 2017.

In that paper, the authors undertake a Mendelian randomisation analysis of CETP genetic variants on cardiovascular disease weighted by their effects on either HDL-C, LDL-C or ApoB. As the reviewer has raised it, we examine the motivation for that paper, the findings, the inferences drawn and what our current paper adds.

The motivation for this analysis, stated in the Introduction to the paper was as follows:

'Although CETP inhibitors were originally designed to increase levels of high-density lipoprotein cholesterol (HDL-C), the more potent CETP inhibitors also robustly lower levels of LDL-C. However, in the ACCELERATE trial, treatment with the CETP inhibitor evacetrapib reduced LDL-C levels by 29mg/dL (0.75 mmol/L) but did not reduce the risk of cardiovascular events. This result has created uncertainty about the causal effect of LDL-C on the risk of cardiovascular disease and raises the possibility that the clinical benefit of lowering LDL-C may depend on how LDL-C is lowered.'

Thus, the Ference study pre-dated the publication of the REVEAL trial of anacetrapib, which was published in NEJM on September 28, 2017. We know from REVEAL trial that potent, specific CETP inhibition for an appropriate duration of follow up leads to a reduction in cardiovascular events. As the reviewer commented earlier (point 4), and as we infer from our current analysis of the totality of evidence, the apparent failure of evacetrapib in a phase 3 clinical trial is as likely to be due to inadequate duration of follow-up as to any difference in the way LDL-C is lowered, or not, via CETP inhibition.

Indeed, this is reflected in the results reported in the Ference paper (our italics):

'In standardized analyses, the CETP score was associated with a very similar risk of major cardiovascular events per 10-mg/dL lower LDL-C level (and per 10-mg/dL lower apoB) as compared with the HMGCR, NPC1L1, and PCSK9 genetic scores. In external replication analyses involving up to 62 240 participants with CHD and 127 299 control participants from the CARDIoGRAMplusC4D Consortium, the CETP score was associated with a lower risk of

CHD (OR, 0.968 [95% CI, 0.956-0.981]; $P < .001$). *This association was very similar in magnitude compared with the association between the HMGCR, NPC1L1, and PCSK9 genetic scores and the risk of CHD per unit change in LDL-C level (and per unit change in apoB).*'

We return to the point about the effect of CETP genetic variants and CETP inhibition on ApoB and LDL-C later.

The analysis conducted by Ference and colleague utilised variants in the CETP gene (acting in *cis*) and examined their association with CVD events. They then undertake an informal comparison of the relationship of the HDL-C, LDL-C and Apo-B altering effects with the magnitude of the effect on CVD. They show that the genetic association with Apo-B provides a stronger (linear) association with CHD, which we do not dispute. This type of analysis is using the lipid or apolipoprotein measure as a proxy for the effect of the genetic variants studied on the level (or function) of the encoded protein. Such a biomarker weighted *cis*-MR analysis should not be confused with a mediation analysis and any inferences drawn about likely mediators should be interpreted with caution.

Our current manuscript takes an importantly different approach to that used by Ference et al. in their 2017 paper. In the current paper we undertake drug target MR analyses of CETP (and PCSK9) also using variants acting in *cis* (since these are less likely to be affected by horizontal pleiotropy) but we analyse their effects on CVD weighted by their effect on circulating CETP and (PCSK9) protein concentration. Since drugs like CETP inhibitors target the encoded protein this type of *cis*-MR analysis weighted by the encoded protein provides the most direct genetic proxy for drug action. Further, we consider the association of CETP and PCSK9 not only with CHD (which we agree was convincingly shown by Ference et al.) but expanded this to consider 190+ pharmacologically important end points, generating several additional important findings for example effects on Alzheimer's disease, age-related macular degeneration, diabetes, heart failure and other end points beyond CHD.

We also undertake additional analyses of the type undertaken by Ference et al. in which we selected genetic variants from the same *cis* region around CETP and used their genetic association with LDL-C, HDL-C, and TG as downstream proxies for their proximal effect on CETP protein concentration.

However, as we describe in our *Nature Communications* paper (Schmidt et al 2020, <https://www.nature.com/articles/s41467-020-16969-0>), drug target MR analyses which use genetic associations with downstream "biomarkers" as proxies for a protein concentration and activity do not provide evidence on whether the biomarker used for the weighting itself *mediates* disease.

To illustrate the argument presented in our 2020 paper, we have included the following diagram, where a genetic variant **G**, the encoded protein **P**, and the downstream biomarker **X** are shown, all of which may have an effect on disease **D**.

Fig. 1 Directed acyclic graphs of potential Mendelian randomisation pathways. Nodes are presented in bold face, with **G** representing a genetic variant, **P** a protein drug target, **X** a biomarker, **D** the outcome, and **U** (potentially unmeasured) common causes of both **P**, **X**, **D**. Labelled paths represent the (causal) effects between nodes.

To see why a biomarker weighted drug target MR (where the genetic effect on **X** is indicated by $\tilde{\delta}\mu$) does not provide evidence for biomarker **X** causing disease **D**, let alone mediation of the **P** \rightarrow **D** pathway, we can consider the case where the biomarker itself does not cause disease; that is when $\theta = 0$.

In this case the only remaining pathway from **P** to disease is ϕ_P , and the “biomarker weighted” drug target MR (ω_{bw}), in the absence of horizontal pleiotropy (that is, when $\phi_G = 0$), is as follows:

$$\begin{aligned}\omega_{bw} &= \frac{\tilde{\delta}(\phi_P + \mu\theta)}{\tilde{\delta}\mu} \\ &= \frac{\phi_P + \mu\theta}{\mu}, (\phi_P \text{ cancels out}) \\ &= \frac{1}{\mu} \times \phi_P \text{ (because } \theta = 0\text{)}\end{aligned}$$

Here we note that the final term involves μ (the protein effect on the biomarker), and ϕ_P (the protein effect on disease), but not θ (the biomarker effect on disease, which is null).

These derivations show that a “biomarker” weighted drug target MR analysis does not provide evidence on the biomarker effect on disease (which can be null). The biomarker is simply providing a proxy for the level of the effect of **G** on **P** and can be useful when **P** is unmeasured. As such it cannot provide evidence on which biomarker mediates the effect of **P** on **D**. As we explain below such mediation *can* however be explored using multivariable MR (MVMR).

Note that none of these arguments disputes the fact that different drug targets that lower LDL-C may do so in conjunction with differing effects on the full lipoprotein and lipoprotein

lipid profile. Indeed, this is the case in the comparison we show in our paper of the effects on the lipoprotein and lipid content profile of CETP and PCSK9 genetic variants. Similar analyses have been undertaken by others:

<https://journals.plos.org/plosbiology/article?id=10.1371/journal.pbio.3000572>.

Moreover, despite our initial aim of specifically assessing the effect of protein inhibition on disease, we agree that some readers might nevertheless be particularly interested in additionally assessing the likely mediating pathway. Hence, also partially based on reviewer 5's suggestion, we employed Multivariable MR (MVMR), which contrary to traditional drug target MR considers multiple biomarkers jointly and in doing so *can* provide evidence on potential mediation pathways. Due to the limited number of variants available in each cis region we developed multiple, two-variable, MVMR models using the available genetic associations with NMR assayed metabolomics (addressing the LDL-C measurement technique commented on below), specifically including LDL-C and HDL-C, and Apo-B and HDL-C.

We therefore clarify these points and include the reference to the Ference 2017 paper on pages 12-14 as follows:

“As well as enabling a separation of on- vs off-target effects of CETP inhibition, drug target MR analyses facilitates investigation of CETP effects beyond those investigated in clinical trials. These drug target MR analyses showed that lower CETP concentration was associated with lower risk of CHD (OR 0.95 per $\mu\text{g/ml}$ lower CETP concentration; 95%CI 0.91; 0.99), HF (OR 0.95; 95%CI 0.92; 0.99) and CKD (OR 0.94; 95%CI 0.91; 0.98), and a higher risk of AMD (OR 1.31; 95%CI 1.22; 1.40). The CETP analyses using genetic associations with LDL-C, HDL-C, or TG, as downstream proxies for lower CETP concentration or activity, additionally suggest that CETP inhibition might protect against T2DM; consistent with findings from CETP-inhibitor trials¹¹. Genetically-instrumented lower PCSK9 concentration was associated with a lower risk of CHD, HF and CKD, and additionally with any stroke, ischemic stroke, AF, MS, as well as an increased risk of asthma and AD¹⁸. Interestingly we did not observe an effect of CETP on AD, despite prior evidence for this¹⁹.

We note that the “biomarker weighted” drug target MR results should not be confused with MR analyses designed to evaluate the causal relevance of major lipid fractions; utilizing genetic variants selected from throughout the genome²⁰. In the presence of post-translation pleiotropy²⁰, where perturbation of a protein could affect multiple downstream biomarkers, some of which may lie on the causal pathway to disease and others not, biomarker weighted drug target MRs, using cis instruments, do not necessarily provide evidence on the possible mediating pathway of the drug target on disease²⁰ and instead reflect drug target effects.

As such our analyses are in line with previous biomarker weighted analyses of CETP and CHD using Apo-B biomarker weights²¹ as proxies for CETP concentration and activity. The current manuscript adds to these results by considering 190 phenotypes including many non-CVD outcomes. We found that CETP and PCSK9 have distinct patterns of effect on the lipoprotein sub-fractions. Both lower genetically predicted CETP and PCSK9 concentration decrease Apo-B, CETP additionally increases HDL-C and Apo-A1 concentration, while decreasing VLDL-C concentration. PCSK9 instead predominantly affects LDL-C sub-

fractions, which are minimally, affected by CETP. This lack of LDL-C effect suggests that the observed LDL-C association in trials are due to reliance on indirect estimation methods, and likely reflects a misclassified VLDL-C effect of CETP. To identify the possible mediating lipid pathway between CETP and disease, we have additionally performed multivariable MR (MVMR) analyses. These analyses suggest that the CHD effect of CETP was (partially) mediated by HDL-C, finding little evidence for LDL-C mediation of the CETP effect on CHD. We similarly identified HDL-C as a likely mediator of the CETP effects on T2DM and AMD. Compared to CETP, the MVMR analysis of PCSK9 convincingly showed LDL-C to be the most likely mediating lipid. Collectively, these findings suggest that, although sharing salutary effects on clinical endpoints, the mechanisms through which CETP and PCSK9 act are likely to be target-specific, as well as outcome-specific. Our findings support the proposal that inhibiting both proteins jointly may elicit benefit, through (multiple) distinct lipid pathways. “

The following was also added on pages 10-11 to reflect the results of the MVMR analysis.

*“Multivariable MR (MVMR) to evaluate lipid mediating pathways
The drug target MR analyses described above used genetic associations with protein concentration, as well as genetic associations with downstream proxies of protein concentration and activity such as LDL-C or HDL-C. Such analyses provide insight on the effects of protein inhibition, but not necessarily on the mediating pathway. To assess such mediation, we employed multivariable MR to jointly model the effects of these lipid pathways, leveraging genetic associations with the more accurate NMR assayed measurements. The MVMR model for LDL-C and HDL-C (Supplemental Figure 4 and Table 7) indicated that the CHD decreasing effects of PCSK9 were convincingly mediated by lower LDL-C (OR per SD decrease in LDL-C: 0.66 95%CI 0.58; 0.75), while for CETP we found this was mediated by HDL-C instead (OR per SD increase in HDL-C: 0.85 95%CI 0.82; 0.88), and not necessarily LDL-C. These analyses further show the risk increasing effect of CETP on AMD was likely due to its HDL-C increasing effect, while the Alzheimer effect of PCSK9 was likely mediated by LDL-C (Supplemental Figure 4). Supplanting LDL-C by genetic associations with Apo-B, we observed suggestive, but insufficiently precise, evidence of Apo-B mediating the CETP effect on CHD OR 0.60 per SD decrease in Apo-B: 95%CI 0.34; 1.03, Supplemental Figure 5 and Table 8). Finally, both MVMR models for CETP indicate its T2DM protective effect acts likely through HDL-C, independent of either LDL-C or Apo-B. “*

6) But maybe , one of the biggest problems with this meta-analysis is the supposed efficacy of the different drugs on LDL-C reductions ; It is listed as -38% for anacetrapib , -37% for evacetrapib etc etc ...These numbers are clearly wrongas Michael Holmes clearly shows in Nature Reviews Cardiology you have to use betaquant to reliably measure LDL lowering with a CETPi and in fact the -41% LDL-C reduction in REVEAL , as assessed by Friedewald or the direct enzymatic assay , is only -17%

One can immediately appreciate that this has to be right when the apoB reduction is considered ; also -18% and completely in line with the LDL-C reduction as assessed by betaquant....

This reviewer has been in touch with Eli Lilly since the LDL-C reductions as reported for ACCELERATE in the NEJM are clearly wrongthe overall apoB reduction in that study is around -19% placebo controlled , but only -15% from baseline whereas the reported LDL-C reduction is -31 %in the M&M is is stated that LDL-C was measured by a direct assay and by betaquant , but the betaquant results are NOT in the NEJM paper , those are the direct assay resultsthe real results will be around 20 % and not morethis is also true for torcetrapib by the way , but is not relevant for dalcetrapibso , all your calculations have to be redone since the numbers you list are a complete over estimation of the real LDL-C lowering effect....only betaquant results are reliable and if not available apoB reductions are the next best.....

Response: We completely agree with the reviewer on the caveats of the source trials relying on the Friedewald equation or beta-quantification to indirectly estimate LDL-C, and further agree that treatment effect estimates of such trials do not necessarily reflect the on-target CETP-inhibitor effects on LDL-C. As the reviewer mentions however, alternative estimates of LDL-C (while sometimes performed) have not been reported in the publicly available literature. However, given our interest in assessing between compound differences, combined with the fact that all studies provide such indirect LDL-C estimations, we find that this does not preclude a meaningful comparator analyses across trials: measurement error induced by relying on the Friedewald equation to estimate LDL-C is consistent across compounds and trials and so will cancel when estimating the difference between compounds in their LDL-C lowering effect. If, hypothetically, our reliance on the Friedewald equation or beta-quantification would favour one particular CETP-inhibitor compound over another, this would predominantly affect LDL-C effect estimates. However, we note that our heterogeneity analysis finds differences between compounds on all the lipid and lipoprotein measurements (including those that have been directly assayed), as well as blood pressure, all-cause mortality, CVD and CHD. Given our aim of testing for the difference between compounds and the fact that the indirect estimations only affects a single outcome (LDL-C), this does not impact the validity of our conclusions.

Additionally, in our MR analyses, we had access to direct (NMR) assays of LDL-C. These analyses indeed show that CETP minimally affects LDL-C, and instead predominantly affects HDL-C, VLDL-C, Apo-A1 and Apo-B.

We discuss this finding, including the reliance of trials and older genetics data on indirect estimation of LDL-C concentration on page 13, and suggest that the observed CETP effect on LDL-C more likely reflects a VLDL-C effect as noted by others:

<https://www.biorxiv.org/content/10.1101/295394v1>.

“We found that CETP and PCSK9 have distinct patterns of effect on the lipoprotein sub-fractions. Both lower genetically predicted CETP and PCSK9 concentration decrease Apo-B, CETP additionally increases HDL-C and Apo-A1 concentration, while decreasing VLDL-C concentration. PCSK9 instead predominantly affects LDL-C sub-fractions, which are minimally, affected by CETP. This lack of LDL-C effect suggests that the observed LDL-C association in trials are due to reliance on indirect estimation methods, and likely reflects a misclassified VLDL-C effect of CETP”

7) In fact , one can even question if dalcetrapib should be part of this analysis at all....in elegant crystallography studies Shenping Liu showed in the J Biol Chem in 2012 that dalcetrapib has a very different chemical structure than any other CETPI , binds to a different pocket on the CETP-protein , does not block the entrance to the tunnel that transports cholesterylesters and binds CETP co-valently to the HDL particlein fact , JTT-705 or dalcetrapib came out of a large scale screen for ACAT inhibitors and was so different to the rest that Roche scientists introduced the term CETP-modulator instead of inhibitor..... It has no effect on apoB , so it will have no effect on MACE

Response: We employed drug target MR weighted by protein concentration to infer on-target effects of CETP inhibition but used compound-specific meta-analyses to formally assess the difference in clinical effects between the various CETP inhibitor compounds. As such we fully agree with the reviewer that dalcetrapib (as well as torcetrapib) shows effects that are distinct from those of a potent specific CETP inhibitor: torcetrapib because of off-target effects and dalcetrapib because of an unusual engagement of the target producing suboptimal inhibition. Both illustrate compound failures but through different mechanisms. Indeed, the inclusion of the dalcetrapib trials unveils this difference and the exclusion of the trials as suggested by the reviewer would obscure it.

Please see page 11

“The reduction in cardiovascular events seen in the REVEAL trial of anacetrapib (median follow-up 1,497 days; Supplementary Table 2) is consistent with the drug target MR results presented here. The manufacturer, Merck, did not seek marketing authorization for this drug, citing anticipated lack of regulatory support¹⁵. The evacetrapib ACCELERATE trial was terminated for futility after a median follow-up of 791 days, a time point before the benefits of anacetrapib emerged in the REVEAL trial (see Figure 1 of ref¹). We further note heterogeneity between compounds. For example, dalcetrapib is a prodrug and modulates rather than competitively inhibits CETP activity and does not affect HDL2 and pre-β HDL subfractions¹⁶. While dalcetrapib likely produced suboptimal CETP inhibition, there is convincing evidence from animal models and human studies that torcetrapib exerts an off-target effect on the adrenal gland through aldosterone that leads to blood pressure elevation, which may have contributed to the increased risk of CVD¹⁷. Nevertheless, it is interesting to note that evacetrapib and anacetrapib both had significant SBP and DBP increasing effects (albeit smaller in magnitude compared to torcetrapib), while dalcetrapib showed a SBP decreasing effect which was in line with the CETP MR results. Taken together, the presented RCT and drug target MR findings, suggest that CETP is a mechanistically viable target for CVD prevention, and the heterogeneous clinical effects of evaluated CETP inhibitors, e.g. the increased risk of mortality and CVD by torcetrapib or the modest lipids effects of dalcetrapib, are likely to be compound - rather than target-related¹⁶. “

8)please consider the clinical update in JAMA 2020 on Lipids and Lipoproteins by Ference et al that conclusively sums up the evidence that apoB is a better risk marker and a better intermediate marker for risk reduction than LDL-C ...I would find it appropriate to use

apoB instead of LDL-C for your paper since for apoB there can be no misunderstanding as to how to measure it.....please realize all data for torcetrapib are Friedewald based and most data for anacetrapib as well as evacetrapib are direct enzymatic assay based...so , the majority of LDL data in your analysis are not correct

Response: Given that the presented meta-analysis of CETP-inhibitors does not involve any decision on Apo-B weighting or mediation, we interpret this comment to refer to our MR analyses. As explained in reply to comment 5, our biomarker weighted MR analyses provide inference on the drug target, that is CETP (or PCSK9), and do not provide information on the potential mediating pathway. Further, in our response to comment 7, we have expanded the revised manuscript to now include multivariable MR analyses which, together with our NMR data, providing further evidence that LDL-C is an unlikely mediator of the CETP effect on CHD.

Please see our replies to both comments 5 and 7 for the revisions made to the manuscript.

9) As a last remark , this reviewer sees no relevance at all for lipoprotein subfractions.....no information beyond apoB has shown to be relevant.....

Response: We respectfully disagree with this comment. These analyses confirm differences in the effect of perturbing CETP and PCSK9 on a wide range of lipid and lipoprotein subfractions and the MVMR analyses we now introduce require these measures to investigate potential mediators.

10) Mendelian randomization

It is quite evident that the apoB lowering effect of CETP-inhibitors must be driven by LDL receptor uptake since CETPi work fine in heFH but not at all in hoFHso it is intuitive why CHD in your MR assay works through LDL-C lowering...what is counterintuitive is the fact that TG lowering (which is irrelevant with CETPi) and HDL-C increase , which was shown in REVEAL to be irrelevant but safe for CHD also lights up in your MR plotsyou need to devote much more discussion to this because here clearly trials are at odds with MR data

Response: As we explained in our response to comment 5 and with reference to our recent Nature Communications paper (Schmidt et al 2020), these biomarker weighted MR analyses proxy protein concentration and activity, and as such do not provide evidence on whether a particular lipid mediates the CETP effect on any specific outcomes. Please see comment 5 for changes made to the manuscript, and additionally we refer to manuscript page 13.

“We note that the “biomarker weighted” drug target MR results should not be confused with MR analyses designed to evaluate the causal relevance of major lipid fractions; utilizing genetic variants selected from throughout the genome²⁰. In the presence of post-translation pleiotropy²⁰, where perturbation of a protein could affect multiple downstream biomarkers, some of which may lie on the causal pathway to disease and others not, biomarker weighted

drug target MRs, using cis instruments, do not necessarily provide evidence on the possible mediating pathway of the drug target on disease²⁰ and instead reflect drug target effects.“

11) What is also incompatible with the evidence is that LDL-C decrease would lower type II DM with CETPiin fact this beneficial effect on HbA1C , HOMA-IR and the incidence of new onset type II is seen with all CETPi's and the best data set is with dalcetrapib , very recently published by Greg Schwartz in Diabetes Care that showed an incredible 26% lower incidence of type II at the end of Dal-Outcomes....dalcetrapib has NO effect on apoB containing lipoproteins , so your analysis cannot be truethe only thing these 4 compounds have in common is the HDL increase and the higher cholesterol efflux capacity....

In fact there is a very sound hypothesis for this mechanism ; removal of toxic sterols from the beta cells in the pancreas by pre-beta HDL , which production is enhanced by at least 30% by CETPi as shown in TULIP in Lancet in 2015 by Hovingh et al

Response: Please see our response to comments 5, 7 and 10 where we explain these analyses do not assess mediation of CETP effects. Additionally, we note that our multivariable MR analyses indeed confirm the trial results cited by the reviewer.

We reflect this as follows:

Page 12

“The CETP analyses using genetic associations with LDL-C, HDL-C, or TG, as downstream proxies for lower CETP concentration or activity additionally suggest that CETP inhibition might protect against T2DM; consistent with findings from CETP-inhibitor trials¹¹.”

Page 13

“We similarly identified HDL-C as a likely mediator of the CETP effects on T2DM and AMD.”

12) This brings me to my last remark ; looking at your HDL plot , the point estimate for protection against type II DM overlaps with the point estimate for protection Alzheimers diseasethere is a revival going on for the sterol hypothesis in AD , see van der Kant et al in Cell Stem Cell , 2019

Quite intriguingly this is based on the lack of efficient cholesterol efflux in apoE4/E4 carriers.....CETPi increase plasma apoE by about 30 to 50% and this apolipoprotein is small enough to cross the BBB at the choroid plexus , so this might be the link between low CETP and protection against Alzheimers which was shown first in centenarian Ashkenazim in New York in JAMA 2015.....

I wonder why you choose to discuss AD for PCSK9 , but not for CETPi ? The biological hypothesis is much stronger for the first than for the latter

Response: Thank you we have added this to the discussion.

“Interestingly we did not observe an effect of CETP on AD, despite prior evidence on this²¹. “

Reviewer #4

Schmidt et al report an interesting meta-analysis on cholesteryl ester transfer protein genes/levels/inhibitors focusing on their association with surrogate cardiovascular and clinical outcomes, and potentially informing on future drug development and adoption. Despite the work strengths, I recommend addressing the following comments:

13). Methods and Results: Test between-agent differences with network or multivariate meta-analysis.

Response: The primary aim of the meta-analysis is to test for between compound differences in their treatment effects compared to placebo in order to ascertain reasons for failure. We have included standard Q-squared tests, showing that the four CETP-inhibitors differ in the clinical effects profile on lipids, blood pressure, as well as clinical outcomes.

Network meta-analyses are most useful in attempting to devise a treatment hierarchy for new agents where each has been evaluated against placebo (or a uniformly used comparison drug) rather than in head-to-head trials. Given that development all four CETP-inhibitors have been terminated and are unavailable for clinical use we have decided against performing a network meta-analysis.

If the reviewer considers that multivariate meta-analyses, jointly testing effects on multiple outcomes, might beneficially guard against multiple testing, such an analysis does not provide details on which outcome is affected by an CETP-inhibitor, nor does it provide direct information on between compound differences (the main aim of the analysis). The included end-point specific estimate would have to be performed in any case.

14). Methods and Results: Appraise with meta-regression if some patient features (eg other conditions, ancillary therapy) impact on the risk-benefit balance of CETP inhibitors.

Response: Thank you for the suggestion. We have performed meta-regression analysis using mean age, mean BMI and proportion of women. Due to the observed heterogeneity between CETP-inhibitors and the implied likelihood of differential off-target effects we have performed these analyses separately for each compound and report the findings in appendix table 4. We find no evidence that these factors explain the observed differences.

15). Methods and Results: Appraise with meta-regression or multivariate meta-analysis which surrogate effects (eg on LDL, CRP or TG) are most predictive of clinical benefits.

Response: Thank you for this suggestion, following the rationale above, we have performed compound-specific meta-regression analyses leveraging treatment effects on the mean difference between baseline change in LDL-C, HDL-C, and Apo-B.

“Due to the limited number of studies available for each compound a meta-regression analyses failed to provide insights into whether the compound specific CVD effects changed by baseline characteristics or lipid effects (Supplemental Table 4).”

16). Methods and Results: Can you identify the optimal dosage to achieve a given effect on surrogate endpoints or on hard clinical outcomes?

Response: We have included results from dose finding studies as supplemental results; page 25. Since development of the four considered compounds has been terminated and optimal dosage is inherently compound specific (as shown the supplemental results), we decided against investigating this further.

17). Methods and Results: Perform detailed small study effect analyses (eg funnel plots, regression tests).

Response: We thank the reviewer for this suggestion and have implemented it.

Page 8

“Small study heterogeneity was explored using funnel plots (Supplemental Figure 2), which did not provide convincing evidence of differential CVD treatment effects by study size, although the number of available studies was limited. Given that the REVEAL anacetrapib trial only showed treatment benefit after the first two years of follow-up¹¹, we analysed short follow-up studies separately: OR 0.80 (95%CI 0.43; 1.48) for CVD, and compared this to REVEAL study CVD estimate: OR: 0.93 (95%CI 0.87; 1.00), showing no significant difference.”

18). Discussion: Elaborate on the most appropriate placement of CETP inhibitors in the context of other lipid lowering therapies, and also on the appropriate per-patient cost of eventually approved agents.

Response: We thank the reviewer for this suggestion and have gladly implemented this focussing on the difference in effects profiles between CETP and PCSK9.

Page 14

“Combined, these findings suggest that, although sharing salutary effects on clinical endpoints, the mechanisms through which CETP and PCSK9 act are likely to be target-specific, as well as outcome-specific. Our findings support the proposal that inhibiting both proteins jointly may elicit benefit, through (multiple) distinct lipid pathways. “

Reviewer #5

This is a very clear and thorough study comparing the effects of HDL-raising therapies with genetic variants that, in part, mimic the effects of those therapies. The results are interesting and provide evidence of potential efficacy and adverse events. The genetic evidence complements and goes beyond the trial evidence because many more conditions can be examined in large existing datasets.

19) The most important comment – i cannot see that the authors have tried to distinguish the HDL-C raising effects from the LDL-C , triglyceride etc lowering effects using the CETP genetic instruments. Weighting the exposure by the main effects on LDL-C, triglycerides and HDL-C etc does not answer the question of what are the marginal effects of these different risk factors because it still focuses on main effects that likely include effects on several pathways (e.g. genetically instrumented lower CEPT alters LDL-C by about a 1/3rd of the effect it raises HDL-C). The authors briefly mention this in the discussion but this is another advantage of the genetic approach and a key question after the on vs off target Q. Is the CETP benefit on CVD outcomes on target and through HDL-C or on-target and through LDL-C, Lipoprotein A , triglycerides etc independent of the other factors – especially independent of LDL-C the strongest risk factor for CVD (although lpA looks to be the most strongly influenced by the genetics of CETP ?). This question can be answered through multi-variable MR. Have the authors considered a multivariable MR ?

Also additional main points:

Response: We agree with the reviewer that this is of interest. In essence there are two issues: (1) on vs off target effects and (2) potential mediators of any beneficial effect on disease end-points. Our main objective was on (1) i.e. to evaluate the phenotypic consequences of inhibiting CETP (or PCSK9) regardless of potential mediating pathways; hence our primary analysis focusing on MR analysis weighted by the effect on the encoded protein. Nevertheless, we agree that additional information on such mediation provides relevant insights, especially for *de-novo* drug development of related lipid-modifying compounds. We have therefore added a multivariable MR analysis (MVMR) which is optimal for exploring potential mediators with the caveat that only a small set of genetic variants (about 15) were available due to the *cis* focus of our analyses. Based on this reviewer's comments and those of reviewer 1, we decided to evaluate an MVMR model with LDL-C and HDL-C, as well as one with Apo-B and LDL-C.

The following is now included in the manuscript.

Page 10

*“Multivariable MR (MVMR) to evaluate lipid mediating pathways
The drug target MR analyses described above used genetic associations with protein concentration, as well as genetic associations with downstream proxies of protein concentration and activity such as LDL-C or HDL-C. Such analyses provide insight on the effects of protein inhibition, but not necessarily on the mediating pathway. To assess such mediation, we employed multivariable MR to jointly model the effects of these lipid pathways, leveraging genetic associations with the more accurate NMR assayed measurements. The MVMR model for LDL-C and HDL-C (Supplemental Figure 4 and Table*

7) indicated that the CHD decreasing effects of PCSK9 were convincingly mediated by lower LDL-C (OR per SD decrease in LDL-C: 0.66 95%CI 0.58; 0.75), while for CETP we found this was mediated by HDL-C instead (OR per SD increase in HDL-C: 0.85 95%CI 0.82; 0.88), and not necessarily LDL-C. These analyses further show the risk increasing effect of CETP on AMD was likely due to its HDL-C increasing effect, while the Alzheimer effect of PCSK9 was likely mediated by LDL-C (Supplemental Figure 4). Supplanting LDL-C by genetic associations with Apo-B, we observed suggestive, but insufficiently precise, evidence of Apo-B mediating the CETP effect on CHD OR 0.60 per SD decrease in Apo-B: 95%CI 0.34; 1.03, Supplemental Figure 5 and Table 8). Finally, both MVMR models for CETP indicate its T2DM protective effect acts likely through HDL-C, independent of either LDL-C or Apo-B. “

Page 13

“To identify the possible mediating lipid pathway between CETP and disease, we have additionally performed preliminary multivariable MR (MVMR) analyses. These analyses suggest that the CHD effect of CETP was (partially) mediated by HDL-C, finding little evidence for LDL-C mediation of the CETP effect on CHD. We similarly identified HDL-C as a likely mediator of the CETP effects on T2DM and AMD. Compared to CETP, the MVMR analysis of PCSK9 convincingly showed LDL-C to be the most likely mediating lipid. Combined, these findings suggest that, although sharing salutary effects on clinical endpoints, the mechanisms through which CETP and PCSK9 act are likely to be target-specific, as well as outcome-specific. Our findings support the proposal that inhibiting both proteins jointly may elicit benefit, through (multiple) distinct lipid pathways.”

Page 17

“We additionally employed multivariable MR (MVMR) to explore potential mediating pathways of the protein (CETP or PCSK9) effects. MVMR evaluates mediation through joint modelling of multiple candidate mediators²⁹, generalizing traditional MR in much the same way as linear regression can be extended to considered multiple (multivariable) risk factors. Due to our focus on the small cis region encoding either PCSK9 or CETP, the number of available variants were limited, diminishing the number of potential mediators we could jointly consider. Hence, we concentrate our analyses on: 1) HDL-C and LDL-C, and 2) HDL-C and Apo-B. Both drug target MR and MVMR invoke a no-horizontal pleiotropy assumption. A drug target MR will be unbiased in the absence of any pre-translation pleiotropy²⁰, but MVMR requires a stronger no-horizontal pleiotropy assumption which additionally assumes all mediators of the protein effect on disease are included in the model. “

20) Figure 5 is a key plot but is not very informative as presented. Why not present the effects and their 95% CIs as a forest plot – the MR estimates scaled to a 1 unit / 1SD increase in the exposure and alongside the actual drug effect ? shades representing p values is not very transparent as to how the effects actually compare. I am not even sure which outcomes each cell refers to.

Response: We appreciate that previous researchers (including ourselves: <https://www.ncbi.nlm.nih.gov/pmc/articles/PMC6820948/>) have often compared drug trial estimates to those obtained from drug target MR analyses, using cross-plots or forest plots. Similar straightforward application is more difficult to apply in the current situation where we do not consider one, but four compounds and one drug target. Illustrations using a common axis is further complicated by the trials using different units than from the genetic data (e.g., percentage change versus mmol/L). We feel this prohibits depicting effect estimates on the same y- or x-axis. Furthermore, while both the meta-analyses of CETP-inhibitors and the MR analysis evaluate CETP inhibition, the effect estimate from a trial refers to a binary variable pertaining to treatment allocation, while MR estimates reflect the effects of a one unit change in a continuous measurement (CETP concentration), hence these estimates are *a priori* expected to differ in magnitude.

We appreciate that this is often addressed by weighting the drug effect estimates and MR estimates by their effect on a specific biomarker (say LDL-C), however as we show in our manuscript CETP affects multiple (lipids) pathways and the choice of biomarker to weight by is therefore difficult, not immaterial, and judging by issues raised by reviewer 1 also not uncontested. In the manuscript (copied below) we discuss these issues and argue that inference on effect direction and evidence against the null-hypothesis (as provided in Figure 5) offers a more robust inferential option. As such, with apologies, we prefer to retain the original figure. We fully agree however, that the figure and its legend should be more self-explanatory and have included an improved description, and also provide guidance how p-values can be mapped to standardized z-statistics and finally we refer the reader to appendix tables which contain all the effect estimates in their native units should be reader prefer a different way of displaying these estimates.

Specific changes:

Figure 5 legend:

“N.B CETP (MR estimates) and drug compound are ordered by columns, with specific outcomes listed in the rows. Effects are presented as $-\log_{10}(p\text{-values}) \times \text{effect direction}$. We note that p-values can be mapped to z-statistics (for example for a p-value of 0.05 we have $-\log_{10}(0.05) = 1.3$, which can be mapped to a z-statistic of 1.96). Clustering was performed on the square root of the $-\log_{10}(p\text{-values}) \times \text{effect direction}$, with the p-value truncated to 10⁻⁶⁰ to ensure sufficient difference between the CETP compound effect on changes in lipids. Associations with a p-value below 0.05 are indicated with a star. The dendrograms represent clustering by outcome (rows) and compound/drug target (columns). Point estimates (OR, MD), confidence intervals and p-values are presented in Supplemental tables 3, 6 and 7.”

Page 14

“Some previous drug target MR studies have attempted to quantify the anticipated effect of a drug targeting the same protein. For example, the anticipated effect of CETP inhibition on CHD risk is a reduction of 40% when weighted by one mmol/L lower LDL-C concentration (Supplemental Figure 3). While of potential interest, there are some caveats that suggest that drug target MR analysis may be more useful as a reliable test of effect direction. This is because drugs that inhibit a target do so usually by modifying its function not its concentration, whereas genetic variants used in MR analysis usually affect protein

expression and therefore concentration. However, for enzymes like CETP, activity reflects both the amount of available protein as well as activity per unit concentration. Thus, on both theoretical grounds and through numerous empirical examples²¹⁻²³, MR analyses using variants in a gene encoding a drug target that affect its expression (or activity) have reproduced the effect direction of compounds with pharmacological action on the same protein²¹⁻²³. Given the typically non-linear drug dose-response, the small downstream effects of genetic variants on the level or function of a protein may underestimate the potential treatment effect of a drug. MR analyses assess the effect of target modulation in any tissue, whereas certain tissues may be inaccessible to a drug either because of its chemistry or the anatomical or physiological barriers. Furthermore, RCTs are closely monitored, and followed for a fixed period, allowing for exploration of induction-times¹¹. MR estimates are considered to reflect a life-long exposure, but in the absence of serial assessment, possible changes across age are difficult to explore, as are disease induction-times. For these reasons we suggest that drug target MR offers a robust indication of effect direction but may not directly anticipate the effect magnitude of pharmacologically interfering with a protein. “

21) I accept that there are caveats and differences between the short term and specific effects of drug action compared with long acting more widespread perhaps effects of genetic variants, but do the authors want to expand on the need to monitor patients on PCSK9 inhibitors for signs of dementia more closely ? this genetic finding , whilst not the primary hypothesis, suggests such a measure might need more follow up work ?

Response: We thank the reviewer for this suggestion which was implemented on page 15.

‘Findings such as the observed increased risk of AMD (from lower CETP), or of asthma Alzheimer’s disease (from lower PCSK9), or the apparent protective effect on MS (from lower PCSK9) provide inference on the likely consequences of protein inhibition. Whether pharmaceutical compounds targeting these proteins elicit similar effects depends on both the duration of drug exposure, as well as the potential for a drug to access the relevant tissues. For example, monoclonal antibody PCSK9 inhibitors may not cross the blood brain barrier. Nevertheless, these findings are relevant for pharmaceutical companies, as well as medicines regulators undertaking post-marketing surveillance of agents targeting these proteins.’

22) Clumping by r^2 of 0.4 sounds too permissive, in that two variants with an r^2 of 0.39 are not providing independent information – indeed they will explain 0.39 of the variance in each other. It is not obvious from the main methods or results how many variants were used for each genetic instrument.

Response: Residual LD was accounted for using reference data from a random sample of 5000 UKB participants and by utilizing MR methods adjusting for between variant correlation. Burgess and colleagues have shown that these methods improve power of localized analyses, such as *cis*-MR, without increasing false positive rates: <https://onlinelibrary.wiley.com/doi/full/10.1002/gepi.22077> . The number of variants used in

each analysis has been included in supplemental tables S5-8, and genomic locations in tables S9-14.

Minor points

23) Why exclude variants < 0.05 for one gene, but not for another ?

Response: We thank the reviewer for pointing this out and have repeated the CETP analysis using a uniform minor allele frequency cut-off of 0.01; results were identical to the prior analyses to 1 decimal place.

Page 16 *“Variants with a minor allele frequency (MAF) below 0.01 were removed”*.

24) In lines 241-245, describing the CETP MR results on major endpoints, it would help to add “lower” and “higher” to the text to help the reader. Likewise abstract lines 70-74

Response: Thank you, implemented as suggested.

25) Line 235 – do you mean Lower “genetically instrumented” CETP concentrations ?

Response: Thank you, we have amended this as suggested.

26) Fig S2 – why is there no data for CETP MR weighted by LDL-C for AMD ?

Response: We apologies for this, the MR estimates for both LDL-C and TG weighted CETP analyses were very extreme, precluding relevant inference – to such an extent that we decided to remove them. We have re-inserted them during revision.

REVIEWER COMMENTS

Reviewer #1 (Remarks to the Author):

dear Authors ,

I am deeply impressed by the scientific rigor of your very extensive responses to my remarks...however , there are 3 important issues still to be adressed that I think you will agree with ...

. page 1

The 2.4 year follow-up of the REVEAL trial was unblinded but still randomizedthe terminal half life of anacetrapib after 4 years of dosing can be measured in years , in fact , it is probably at least 4 years...so , patients randomized to anacetrapib were still on the full dose of the drug for the 2.4 years of follow-up....and patients randomized to placebo were still on nothing 2.4 years after the trial

This provided the unique opportunity to study efficacy as well as safety of anacetrapib over an unusually long period of time...moreover , also the 6.4 year follow up results for MACE were on the CTT meta-regression line , indicating non-HDL, apoB or LDL-C to be the mediators of effect.....the absolute LDL-C reduction , measured in a subgroup of REVEAL by betaquant , was in fact only 11 mg/dl and precisely predicted the efficacy in terms of MACE after 6.4 years of therapy

This a very important point that ties in directly with the next issue

Page 9

"This lack of LDL-C effect suggests that the observed LDL-C association in trials is due to reliance on indirect estimation methods and likely reflects a misclassified VLDL-C effect of CETPi ."

This is clearly wrong ; please look at the TULIP results with obicetrapib , in which ALL lipoprotein measurements were done by ultracentrifugation ; LDL-C reductions up to 45.3%and lets make no mistake ; betaquant is the gold standard and not NMR , a method that is increasingly questioned in the lipid arena....

Page 10

"Alternative estimates of LDL-C have not been reported in the publicly available litterature " This statement is incorrect , LDL-C levels are measured by betaquant for obicetrapib as well as for

anacetrapib....in a subset of 2000 participants betaquant was used to determine LDL-C and the reduction was -17% with an absolute reduction of just 11mg/dl

I can live with leaving the 6.4 year follow -up of REVEAL out of the equation , but I cannot live with letting the readership think that CETPi lowers VLDL-C whereas there is rock solid evidence to the contrary and obtained by using the gold standard for LDL-C measurement...

Reviewer #4 (Remarks to the Author):

All my comments have been reasonably addressed.

Reviewer #5 (Remarks to the Author):

Many thanks for performing the MVMR. I realise teasing apart the HDL-C from non-HDL-C effects were not the primary hypothesis, but the mediating role is a key part of this story I believe. It is interesting that the MR-instrumented HDL-C effects of CETP inhibition adjusted for LDL-C on CHD seem to be stronger than the MR-instrumented LDL-C effects of PCSK9 inhibition adjusted for HDL-C. That seems a little strange given LDL-C has such a stronger role.

I also think the adjustment for APOB in the MVMR is key. I'd encourage the authors to include supp figures 4 and 5 in the main paper if they and editors can find room, because if low HDL-C is causal it is (I believe) a seriously important result.

Tim Frayling

Reviewer #1

dear Authors ,

I am deeply impressed by the scientific rigor of your very extensive responses to my remarks...however , there are 3 important issues still to be adressed that I think you will agree with ...

The 2.4 year follow-up of the REVEAL trial was unblinded but still randomizedthe terminal half life of anacetrapib after 4 years of dosing can be measured in years , in fact , it is probably at least 4 years...so , patients randomized to anacetrapib were still on the full dose of the drug for the 2.4 years of follow-up....and patients randomized to placebo were still on nothing 2.4 years after the trial

This provided the unique opportunity to study efficacy as well as safety of anacetrapib over an unusually long period of time...moreover , also the 6.4 year follow up results for MACE were on the CTT meta-regression line , indicating non-HDL, apoB or LDL-C to be the mediators of effect.....the absolute LDL-C reduction , measured in a subgroup of REVEAL by betaquant , was in fact only 11 mg/dl and precisely predicted the efficacy in terms of MACE after 6.4 years of therapy

This a very important point that ties in directly with the next issue

Response: We thank the reviewer for allowing us to provide further explanation on why we did not include the unpublished post-trial data from the unblinded follow-up of the REVEAL study, and instead focused on the peer-reviewed REVEAL data from the blinded follow-up period.

While randomization ensures patient groups are similar *at baseline*, blinding ensures they act similarly *during follow-up*. Hence without sufficient blinding, especially due to the duration of the extended unblinded follow-up, there is no guarantee that these effects can be robustly ascribed to the allocated study interventions (irrespective of randomization); for the relevant statistical derivation, see Schmidt SMMR 2016: <https://pubmed.ncbi.nlm.nih.gov/27932664/>.

Our compound-specific meta-analysis aims to leverage between compound heterogeneity as a measure to distinguish between failures attributable to the drug target (CETP) and failures related to properties of the individual inhibitor compounds. Including data from an unblinded phase of a trial risks inducing heterogeneity which does not relate to the study drug, but instead to the study design. As such, while we agree these results are important and deserve mention, we feel including these data in our meta-analysis would weaken our interpretation of differences between compounds, by introducing a new, potentially confounding variable of difference in trial design. Nevertheless, we strongly agree readers of our manuscript should be made aware of the results of the longer term unblinded follow up. As such, we have expanded the discussion as follows on page 10:

“The evacetrapib ACCELERATE trial was terminated for futility after a median follow-up of 791 days, a time point before the benefits of anacetrapib emerged in the REVEAL trial (see Figure 1 of ref¹); the anacetrapib effect on major coronary events increased to a rate ratio of 0.80 (95%CI 0.71; 0.90) during an additional median 2.3 years unblinded follow-up. “

Regarding the outstanding question on which lipid particle, or particles, mediate the CETP effect on CHD. We remind the reviewer that the primary aim of our contribution is to determine the *on-target* effects on disease endpoints of CETP inhibition, not the mediator(s) of this effect. While we agree that elucidating the *on-target* mediation pathway of CETP might help inform priority given to novel targets, for example to see if they share similar mediating pathways, the issue of which mediator is less relevant for attempts to drug CETP – as the mediation pathway for a specific CETP-inhibitor with no off-target effects should be the same as that instrumented genetically through the protein.

The reviewer suggests that the, as yet unpublished, post-trial follow-up data from the REVEAL study provides some insights on this mediation question. We note that, as far as we can tell, these results have only been published as PowerPoint slides, which did not include any meta-regression line:

https://www.revealtrial.org/REVEAL_AHA_PTFU_Slides_2019_11_12.pdf

Additionally, the reviewer refers to the “CTT meta-regression”, due to the absence of a meta-regression line in the provided link this is of course difficult to evaluate, however we can only assume that “CTT” refers to the Cholesterol Treatment-Trialists (CTT) meta-analysis of statin trials, not CETP inhibitors. Despite not being able to access these data, it seems unlikely that a statin meta-regression line (which relates to a drug class targeting HMGCR rather than CETP) would provide *direct* evidence on the mediating pathway of CETP itself. Instead, our contribution already includes meta-regression analyses which rely on compound-specific CETP inhibitor effects, not statin data. Additionally, we note that meta-regression makes observations on the aggregate level, where a single trial maps to a single observations. Such meta-regression analyses are inherently observational and subject to bias (Thompson and Sharp in StatMed 1999), and hence do not provide the same level of evidence as the source trials themselves.

In summary, we do not pursue this issue further because: 1) We have been unable to find the data the reviewer mentioned, 2) it seems to pertain to statin trial data and not to CETP, 3) the analysis method (meta-regression) is subject to bias, 4) we have already included a meta-regression analysis of CETP compounds in the current manuscript, and 5) the question on mediation of the CETP to CHD effect deviates from the primary aim of our paper: To assess the on-target effects of CETP and the validity of CETP as a drug target for CVD.

"This lack of LDL-C effect suggests that the observed LDL-C association in trials is due to reliance on indirect estimation methods and likely reflects a misclassified VLDL-C effect of CETPi ."

This is clearly wrong ; please look at the TULIP results with obicetrapib , in which ALL lipoprotein measurements were done by ultracentrifugation ; LDL-C reductions up to 45.3%and lets make no mistake ; betaquant is the gold standard and not NMR , a method that is increasingly questioned in the lipid arena....

Response: We thank the reviewer for allowing us to improve this section. We note that our findings pertaining to a VLDL-C effect of CETP, inferred through genomics, were also shown

by Blauw et al, and by Kettunen et al, both based on NMR spectroscopy, and referred to in the Nat Rev Cardiology reference (no. 22) from Holmes and Ala-Korpela 2019, suggested by the reviewer during the first revision. We have therefore rewritten the relevant section as follows:

"We note that in the MR analysis, CETP concentration was only minimally associated with LDL-C measured by NMR. This was distinct from a strong apparent cis-MR CETP effect on LDL-C using data from GLGC, which indirectly measured LDL-C using the Friedwald equation (which may misclassify VLDL-C and IDL-C as LDL-C²²), or the beta-quantification method (which may misclassify IDL-C as LDL-C²²). In general lipoprotein subclass-specific methods such as provided by NMR spectroscopy represent a more accurate measurement of LDL-C²². The accuracy of NMR-assayed LDL-C was empirically validated in PCSK9 MR analysis which showed a strong (and specific) association with LDL-C (Figure 4). Additionally, we note that the absence of a LDL-C association by CETP has been reported before by Blauw et al²³, and by Kettunen et al²⁴, both based on NMR spectroscopy. While the above provides evidence against a meaningful LDL-C effect of CETP, when LDL-C is measured accurately and specifically, a more definitive answer on its relation to pharmacologically inhibiting CETP requires further consideration. This is particularly important given 1) the heterogeneous CETP inhibitor effects of different compounds (Figure 1; e.g., dalcetrapib minimally affected LDL-C concentrations), 2) the absence of CETP inhibitors trial data using similar lipoprotein subclass-specific methods to directly assay LDL-C concentrations, and 3) the absence of trial data on possible VLDL-C and IDL-C effects of CETP inhibition."

"Alternative estimates of LDL-C have not been reported in the publicly available literature " This statement is incorrect , LDL-C levels are measured by betaquant for obicetrapib as well as for anacetrapib....in a subset of 2000 participants betaquant was used to determine LDL-C and the reduction was -17% with an absolute reduction of just 11mg/dl

Response: We agree and wish to explain that we intended to merely acknowledge a comment made by the reviewer, that the *majority* of the available LDL-C measurements in the trial arena were made using the Friedewald equation. We note that the quoted text pertains to our rebuttal letter, not to the manuscript text. We additionally confirmed the manuscript text did not contain phrasing that might mislead the reader, so we have made no amendments to the manuscript in relation to this comment.

Finally, quoting the previous rebuttal letter, we remind the reviewer that the discussion on the quality of the LDL-C assay, while important, pertains to a single phenotype, and does not impact our analyses of the remaining lipid measurements, blood pressure and other physiological measures and clinical endpoints.

"If, hypothetically, our reliance on the Friedewald equation or beta-quantification would favour one particular CETP-inhibitor compound over another, this would predominantly affect LDL-C effect estimates. However, we note that our heterogeneity analysis finds differences between compounds on all the lipid and lipoprotein measurements (including those that have been directly assayed), as well as blood pressure, all-cause mortality, CVD

and CHD. Given our aim of testing for the difference between compounds and the fact that the indirect estimations only affect a single outcome (LDL-C), this does not impact the validity of our conclusions.”

I can live with leaving the 6.4 year follow -up of REVEAL out of the equation , but I cannot live with letting the readership think that CETPi lowers VLDL-C whereas there is rock solid evidence to the contrary and obtained by using the gold standard for LDL-C measurement...

Response: We are grateful to the reviewer for sharing their experience on this very important topic, which we feel has further improved our manuscript, especially for a more clinically orientated readership.

We note that none of the CETP inhibitor trials have previously reported on VLDL-C or IDL-C, and hence provide no evidence against an association with these lipid measurements. Additionally, given the role that CETP plays in the lipid metabolism, there is little reason to doubt the plausibility of an association of CETP with these lipid fractions. We also re-iterate that our findings pertaining to a VLDL-C effect of CETP inhibition, inferred through genomics, were also shown by Blauw et al, and by Kettunen at al, both based on NMR spectroscopy, and referred to in the Nat Rev Cardiology reference (no. 22) from Holmes and Ala-Korpela 2019, suggested by the reviewer during the first revision, page 12.

Reviewer #4 (Remarks to the Author):

All my comments have been reasonably addressed.

Response: We wish to thank the reviewer for their valuable comments.

Reviewer #5 (Remarks to the Author):

Many thanks for performing the MVMR. I realise teasing apart the HDL-C from non-HDL-C effects were not the primary hypothesis, but the mediating role is a key part of this story I believe. It is interesting that the MR-instrumented HDL-C effects of CETP inhibition adjusted for LDL-C on CHD seem to be stronger than the MR-instrumented LDL-C effects of PCSK9 inhibition adjusted for HDL-C. That seems a little strange given LDL-C has such a stronger role.

Response: We agree and are grateful for the reviewer for suggesting these.

Regarding the current analyses, the CETP effect of *higher* HDL-C on CHD conditional on LDL-C is OR 0.85 (95%CI 0.82; 0.88), compared to the PCSK9 effect of *lower* LDL-C on CHD conditional on HDL-C of OR 0.66 (95%CI 0.58;0.75); see Table S7. As such the effect magnitude of CETP for HDL-C is *smaller* than that of PCKS9 for LDL-C.

I also think the adjustment for APOB in the MVMR is key. I'd encourage the authors to include supp figures 4 and 5 in the main paper if they and editors can find room, because if low HDL-C is causal it is (I believe) a seriously important result.

Response: Thank you, we have included APO-B adjusted results in Table S8 and supplemental Figure S4. As suggested, we have added the HDL-C and LDL-C MVMR figure to the main manuscript (Figure 6).

REVIEWER COMMENTS

Reviewer #1 (Remarks to the Author):

dear authors ,

it is safe to assume that the CTT meta-regression line is indeed the line described by the Oxford group that has established the relationship between absolute changes in LDL-C , non-HDL-C or apoB and reductions in MACEthat line , for non-HDL , is published in the Supplementary appendix of the NEJM REVEAL publication , Figure S5the absolute non-HDL reduction in REVEAL falls exactly on the statin meta-regression line ; a very powerful argument in favour of the contention that CETP-inhibition lowers MACE risk through lowering of LDL-C or non-HDL-C....and that line is not "subject to bias " as the authors incorrectly claim , but instead forms the foundation of the claim in the EAS/ESC and AHA/ACC guidelines that the LDL hypothesis no longer exists but is rather an axiom .

The next point is even more important ; the authors refuse to concede to 2 objections I have made earlier ;

preparative ultracentrifugation or betaquant is the Gold Standard for measuring LDL-c ...the number of papers stating this are too many to reference here , please look at Paul Hopkins paper in Atherosclerosis 243(2015)99-106 or for that matter the ACC position statement in 2020 by Roger Blumenthal.

NMR is not even the gold standard for measuring particle numbers

And with this Gold Standard it is crystal clear that potent CETP-inhibitors like anacetrapib and obicetrapib have the largest effect on LDL-c levels and some effect on VLDL-C levelsand both data are published in very important Journals . IN the Lancet paper from 2015 obicetrapib lowered LDL-C by 45.3% and VLDL-C by 17% .1 ..

And so the entire section from "We note ..."contains factual errors ...

. NMR does NOT represent a more accurate measurement of LDL-C

.Blauw and Kettunen , like the authors of this paper , came to the WRONG conclusion because they based it on NMR and not on betaquant

. there is NO absence of trial data ; Lancet 2015 , 364 patients , all assessed by betaquant...clearly the largest effect on LDL-C , and a much smaller effect on VLDL-C ...

So , my point is that trial data , published in Lancet in 2015 , show that a potent CETP-inhibitor mainly lowers LDL-C and to a lesser degree VLDL-C , my next point is that NMR is inferior to betaquant for LDL-C assessment and my last point is that anacetrapib , also shown by betaquant to mainly lower LDL-C , in a 30.000 patient CVOT , has produced a MACE result that falls exactly on the non-HDL line

Reviewer #1

dear authors ,

it is safe to assume that the CTT meta-regression line is indeed the line described by the Oxford group that has established the relationship between absolute changes in LDL-C , non-HDL-C or apoB and reductions in MACEthat line , for non-HDL , is published in the Supplementary appendix of the NEJM REVEAL publication , Figure S5the absolute non-HDL reduction in REVEAL falls exactly on the statin meta-regression line ; a very powerful argument in favour of the contention that CETP-inhibition lowers MACE risk through lowering of LDL-C or non-HDL-C....and that line is not "subject to bias " as the authors incorrectly claim , but instead forms the foundation of the claim in the EAS/ESC and AHA/ACC guidelines that the LDL hypothesis no longer exists but is rather an axiom .

The next point is even more important ; the authors refuse to concede to 2 objections I have made earlier ;

preparative ultracentrifugation or betaquant is the Gold Standard for measuring LDL-c ...the number of papers stating this are too many to reference here , please look at Paul Hopkins paper in Atherosclerosis 243(2015)99-106 or for that matter the ACC position statement in 2020 by Roger Blumenthal.

NMR is not even the gold standard for measuring particle numbers

And with this Gold Standard it is crystal clear that potent CETP-inhibitors like anacetrapib and obicetrapib have the largest effect on LDL-c levels and some effect on VLDL-C levelsand both data are published in very important Journals . IN the Lancet paper from 2015 obicetrapib lowered LDL-C by 45.3% and VLDL-C by 17% .1 ..

And so the entire section from "We note ..."contains factual errors ...

. NMR does NOT represent a more accurate measurement of LDL-C

.Blauw and Kettunen , like the authors of this paper , came to the WRONG conclusion because they based it on NMR and not on betaquant

. there is NO absence of trial data ; Lancet 2015 , 364 patients , all assessed by betaquant...clearly the largest effect on LDL-C , and a much smaller effect on VLDL-C ...

So , my point is that trial data , published in Lancet in 2015 , show that a potent CETP-inhibitor mainly lowers LDL-C and to a lesser degree VLDL-C , my next point is that NMR is inferior to betaquant for LDL-C assessment and my last point is that anacetrapib , also shown by betaquant to mainly lower LDL-C , in a 30.000 patient CVOT , has produced a MACE result that falls exactly on the non-HDL line

Response: We thank the reviewer for the detailed guidance, which we summarize as follows.

University College London, Gower Street, London WC1E 6BT

Tel: 0044 (0)20 3549 5625

amand.schmidt@ucl.ac.uk

www.ucl.ac.uk

The reviewer proposes that a reduction in LDL-C (i.e. the cholesterol content of low-density lipoprotein particles, rather than cholesterol in other particles, e.g. VLDL) that is the likely mediator of the beneficial effect of CETP inhibition on CHD. This is based on the following lines of reasoning:

1. In trials of specific, potent CETP inhibitor drugs, e.g. obicetrapib and anacetrapib, treatment with the active agent vs. placebo leads to a lowering of LDL-C, as measured by methods other than NMR-spectroscopy.
2. The reduction in CHD events in the REVEAL trial of anacetrapib (a specific, potent CETP-inhibitor) appears to be in proportion to its lowering of non-HDL-C (which is calculated as total cholesterol minus HDL-C, measured by methods other than NMR-spectroscopy), where the relationship between non-HDL-C lowering and reduction in CHD event rate is set in the context of the prior trials of statins, as summarised by the Cholesterol Treatment Trialists Collaboration.
3. Preparative 'ultracentrifugation or betaquant' is the gold-standard for measurement of LDL-C citing a paper by Hopkins et al. *Atherosclerosis* 243(2015)99-106.

The editors propose a 'significantly revised presentation/discussion of the results pertaining to LDL-C, with particular attention to the discussion section starting with „ We note that in the MR analysis, CETP concentration was only minimally associated with...". Specifically, the editors request 'inclusion/discussion of the three main points highlighted by reviewer #1, i.e. REVEAL results falling exactly on CTT meta-regression line (and what this implies, as mentioned by the reviewer), previous trials/studies showing LDL-C reductions by betaquant, and relative accuracy of NMR vs betaquant (citing relevant references such as <https://pubmed.ncbi.nlm.nih.gov/26363807/>). The editors request 'discussion of these points with appropriate references throughout. The scope of such revision should be to provide a more balanced and careful discussion of the available evidence that CETP inhibitors lead to lower LDL-C acknowledging that limitations of NMR as a quantifying method for LDL-C could potentially affect the conclusions of your work on LDL-C changes/mediating pathways.'

In revising the cited section, we have taken the following into consideration:

Reviewer point 1: We agree it is clear that specific, potent CETP inhibitors lead to a reduction in LDL-C when measured using methods other than NMR-spectroscopy. We discuss the likely explanation for this under Reviewer point 3 below.

Reviewer point 2: Although the reduction in coronary death and MI in the REVEAL trial of anacetrapib is consistent with the degree of non-HDL-C lowering, when set in the context of trials of statin drugs, the authors of the REVEAL trial note "It is not possible to determine the mechanism by which anacetrapib reduced the risk of major coronary events in this trial." There are two reasons for this. First, an evaluation of the relationship between a variable (here non-HDL-C) and outcome (here CHD events) across trials (rather than across randomized intervention and placebo arms) is necessarily observational, not randomised and cannot prove a cause-effect relationship. Second, such a univariable analysis is identical to a simple (weighted) Pearson correlation of the standardized dependent and independent variables. As such, a univariable meta-regression analysis can only provide information on mediation if one is willing to assume that aggregated studies were identical in

setting, patient characteristics and that the compound only affects disease through the single variable that is plotted on the x-axis. Given that CETP inhibition is known to affect more than one lipid and lipoprotein subfraction as well as apolipoprotein concentration, such a univariable meta-regression analyses cannot exclude the possibility that some of these closely related variables may instead explain the linear trend and hence represent the true mediation pathway or, alternatively, that there may be multiple mediating pathways towards disease.

We do agree, however, that the close “fit” of the REVEAL study with the CTT estimates is important information to refer to and comment on.

Reviewer point 3: Here it is important to note that LDL-C measured using NMR spectroscopy is a measure of the cholesterol content of low-density lipoproteins defined by particle size: i.e. a size-specific LDL-C measure. The more common estimation of LDL-C by the Friedwald equation includes the measurement of the cholesterol content of IDL, VLDL, and lipoprotein(a), which do not contribute to the size-specific LDL-C measurement when made using NMR spectroscopy. Thus, the comparison of LDL-C estimated using the Friedwald equation and directly measured by NMR-spectroscopy is not a like-for-like comparison. Likewise, non-HDL-C (which was measured in the REVEAL trial) is defined as total cholesterol (i.e. the cholesterol on all lipoprotein subfractions) *minus* the cholesterol in high-density lipoproteins. Non-HDL-C therefore includes the measurement of the cholesterol content of VLDL, IDL and LDL. Regarding beta quantification, as argued by Holmes and Ala-Korpela in their *Nature Reviews Cardiology* article entitled “What is LDL-C?: “Beta-quantification involves ultracentrifugation to remove chylomicrons and VLDL particles by density (leading to a so-called bottom fraction) and then precipitation of the apoB-containing particles to measure HDL cholesterol and obtain the level of ‘LDL cholesterol’ by subtracting HDL-cholesterol levels from the cholesterol in the bottom fraction. Therefore, beta-quantified ‘LDL cholesterol’ also includes IDL cholesterol in the reported ‘LDL-cholesterol. Indeed, only lipoprotein subclass-specific methods can accurately quantify cholesterol in LDL particles, which can be achieved by size-specific quantification of lipoproteins via high-performance liquid chromatography (HPLC) or nuclear magnetic resonance (NMR) spectroscopy or via density-specific ultracentrifugation.” Therefore, the point we make is that there is a difference in the pattern of effect of PCSK9 and CETP, instrumented genetically, on the cholesterol content of VLDL, IDL and LDL that is masked when LDL-C is estimated using the Friedwald method or beta quantification rather than directly measured using NMR spectroscopy. Importantly, a similar discrepancy has been noted in a comparison of the effect of variants in the CETP and HMGCR genes on the cholesterol content of various size-specific lipoprotein fractions measured by NMR spectroscopy in Finnish cohorts and the INTERVAL study, as reported by Kettunen et al. (*PLoS Biology*, 2019).

The paper by Blauw and Kettunen to which the reviewer refers (*European Journal of Human Genetics* (2019) 27:422–431 <https://doi.org/10.1038/s41431-018-0301-5>) is based on a Mendelian randomisation analysis of circulating lipoproteins measured using the Nightingale platform in 5672 samples from the NEO cohort. Again, entirely consistent with our current findings, the authors of this study concluded, based on their genetic analysis that: “CETP inhibition causes a relatively large increase in HDL components, which is predominantly caused by an increase in large and medium-sized HDL particles, in addition to a more modest reduction in VLDL components, which is mainly caused by a decrease in small and

extra small VLDL particles. Of note, LDL concentration and composition were not affected by CETP, which was unexpected based on the current dogma that CETP increases LDL-C.

Therefore, in three very large studies (including our own), across multiple cohorts, there are consistent findings on the discrepant effects of CETP on size-specific vs non-size specific measurement of LDL-C, and a consistent effect on size-specific HDL and VLDL, with comparatively a small effect on LDL.

The reference provided by the reviewer (Hopkins et al. *Atherosclerosis* 243(2015)99-106) tests agreement between four methods including two NMR based methods. However, the NMR methods are from Liposcience and Health Diagnostic Laboratory but not the NMR method used here (from Nightingale Laboratories). As shown in figure 2 from Würtz and Soininen [https://linkinghub.elsevier.com/retrieve/pii/S0021-9150\(20\)30193-3](https://linkinghub.elsevier.com/retrieve/pii/S0021-9150(20)30193-3) there is a strong correlation (0.88 or larger) between size specific LDL-C subfractions measured using HPLC and Nightingale NMR, confirming the accuracy of *Nightingale* NMR platform for the estimation of LDL-C. Additionally, in the Tikkanen et al 2020 medRxiv paper, specifically supplemental figure 1, (<https://www.medrxiv.org/content/10.1101/2020.07.24.20158675v1>) there is empirical data to support the argument proposed by Holmes and Ala-Korpela referred to above. Here, Tikkanen et al show (across five independent cohort studies) that there is a strong correlation between clinical chemistry measured LDL-C and a Nightingale “clinical” LDL-C (that incorporates the additional lipoprotein subfractions included in the clinical chemistry methods).

Therefore, as directed by the editors, we have taken the reviewer’s viewpoint and these various lines of evidence to produce a balanced discussion in the paper, rewriting the key section of the Discussion as follows:

Page 12-13

“As described above, the *cis*-MR analysis weighted by CETP concentration indicated a causal effect of lower CETP concentration on HDL-C, VLDL-C, IDL-C, as well as Apo-A1 and Apo-B, but only to a lesser extent with LDL-C measured by NMR spectroscopy using the Nightingale platform. This finding is consistent with the findings of Blauw et al in 5672 participants from the NEO study²¹ and of Kettunen et al., in Finnish cohorts and the INTERVAL study²². However, we did identify a strong *cis*-MR CETP effect on LDL-C when LDL-C was measured using clinical chemistry methods. Notably, by contrast, our MR analysis of the effect of PCSK9 on lipids and lipoproteins showed the expected association with LDL-C assayed both by non-size specific methods and by the Nightingale-NMR platform. A potential explanation for this discrepancy may be found in Tikkanen et al. who identified a strong correlation between clinical chemistry measured LDL-C and a derived “clinical” LDL-measure from the Nightingale NMR platform²³ that incorporates the additional lipoprotein subfractions VLDL-C, IDL-C and lipoprotein(a), which are included as part of the clinical chemistry based assay methods). However, other NMR-based methods from Liposcience and Health Diagnostic Laboratory have shown limited agreement with other methods in the measurement of LDL-C²⁴.

We note that in the REVEAL trial of anacetrapib and the TULIP trial of obicetrapib, these specific, potent CETP inhibitors consistently show an LDL-C lowering effect (when LDL-C is measured using non-NMR based methods and that the reduction in CHD events in the REVEAL trial of anacetrapib (a specific, potent CETP-inhibitor) appears to be in proportion to its lowering of non-HDL-C, where the relationship between non-HDL-C lowering and reduction in CHD event rate is set in the context of the prior trials of statins, as summarised by the Cholesterol Treatment Trialists collaboration (see Figure

5S in ref¹¹). To further clarify the explanation for these discrepancies, it would be important to perform analyses using both size-specific (NMR) and non-size specific LDL-C assay methods in the same trial participants.”